# ADAPTING TO LABEL SHIFT WITH BIAS-CORRECTED CALIBRATION

## ABSTRACT

Label shift refers to the phenomenon where the marginal probability $p(y)$ of observing a particular class changes between the training and test distributions, while the conditional probability $p(\boldsymbol{x}|y)$ stays fixed. This is relevant in settings such as medical diagnosis, where a classifier trained to predict disease based on observed symptoms may need to be adapted to a different distribution where the baseline frequency of the disease is higher. Given estimates of $p(y|\boldsymbol{x})$ from a predictive model, Saerens et al. (2002) proposed an Expectation Maximization (EM) algorithm to efficiently correct for the difference in class proportions between training and test distributions. A limiting assumption of this EM algorithm is that the probabilities $p(y|\boldsymbol{x})$ are calibrated, which is not true of modern neural networks. Recently, Black Box Shift Learning (BBSL) (Lipton et al., 2018) and Regularized Learning under Label Shifts (RLLS) (Azizzadenesheli et al., 2019) have emerged as state-of-the-art techniques to cope with label shift when a classifier does not output calibrated probabilities. In this work, we show that while EM in the absence of calibration achieves weak performance with modern neural networks, the combination of EM with appropriate calibration typically outperforms BBSL and RLLS. In particular, we show that the popular calibration strategy of Temperature Scaling (TS) is often insufficient to completely correct miscalibration; rather, EM achieve state-of-the-art results primarily when used with a calibration approach that includes class-specific bias parameters that TS lacks. We further introduce a theoretically principled strategy for estimating source-domain priors in EM-based domain adaptation that allows the EM algorithm to behave more robustly when the calibrated probabilities contain systematic bias. Colab notebooks reproducing experiments on CIFAR10, CIFAR100, MNIST and diabetic retinopathy detection are available at (anonymized link): `https://github.com/blindauth/labelshiftexperiments`

## 1 INTRODUCTION

Imagine we train a classifier in country A to predict whether or not a person has a disease based on observed symptoms, and that we hope to deploy this classifier in country B, which has poorer access to healthcare. If the prevalence of the disease in country B is higher in than in country A, the classifier might systematically misdiagnose people as not having the disease. How can we adapt the classifier to cope with the difference in the baseline prevalence of the disease in the two countries?

Formally, let $y$ denote our labels (e.g. whether or not a person is diseased), and let $\boldsymbol{x}$ denote the observed symptoms. Let us denote the joint distribution $(\boldsymbol{x}, y)$ in country A (our "source" domain) as $\mathbb{P}$, and let us denote the distribution in country B (our "target" domain, where we do not have labels) as $\mathbb{Q}$. How can we adapt a classifier trained to estimate $p(y|\boldsymbol{x})$ (the conditional probability in distribution $\mathbb{P}$) so that it can instead estimate $q(y|\boldsymbol{x})$ (the conditional probability in distribution $\mathbb{Q}$)? Absent assumptions about the nature of the shift between $\mathbb{P}$ and $\mathbb{Q}$, this problem is intractable. However, if the disease generates similar symptoms in both countries, we can assume that $p(\boldsymbol{x}|y) = q(\boldsymbol{x}|y)$, and that the shift in the joint distribution $q(\boldsymbol{x}, y)$ is due to a shift in the label proportion $q(y)$. Formally, we assume that $q(\boldsymbol{x}, y) = p(\boldsymbol{x}|y)q(y)$. This is known as *label shift* or *prior probability shift* (Amos, 2008), and it corresponds to anti-causal learning (i.e. predicting the cause $y$ from its effects $\boldsymbol{x}$) (Schoelkopf et al., 2012). Anti-causal learning is appropriate for diagnosing diseases given observations of symptoms because diseases cause symptoms.

Given estimates of $p(y)$ and $p(y|\boldsymbol{x})$, Saerens et al. (2002) proposed a simple Expectation Maximization (EM) procedure to estimate $q(y)$ without needing to estimate $p(\boldsymbol{x}|y)$. However, estimates of $p(y|\boldsymbol{x})$ derived from modern neural networks are often poorly calibrated (Guo et al., 2017), and the lack of calibration can decrease the effectiveness of EM. As an alternative, Lipton et al. (2018) developed a technique called Black Box Shift Learning (BBSL) that can work even when the predictions $p(\boldsymbol{x}|y)$ are not calibrated. Azizzadenesheli et al. (2019) further improved upon BBSL in a technique known as Regularized Learning under Label Shifts (RLLS). Both BBSL and RLLS leverage information in a confusion matrix calculated on a held-out portion of the training set. To our knowledge, neither BBSL and RLLS have been benchmarked against the EM approach.

Although the EM approach is limited by the assumption that the predictions $p(y|\boldsymbol{x})$ are calibrated, a number of recent techniques have been proposed to correct for miscalibration of $p(y|\boldsymbol{x})$ using a held-out portion of the training set (Guo et al., 2017). The held-out set can be thought of as analogous to the held-out set used in BBSL and RLLS to calculate a confusion matrix. This suggests a simple yet novel hybrid algorithm for adapting to label shift: first, calibrate predictions using the held-out training set, then perform domain adaptation on the calibrated predictions using EM. In this work, we studied the effectiveness of this hybrid algorithm. More generally, we studied the impact of calibration on domain adaptation to label shift. Our primary contributions are as follows:

## 1.1 OUR CONTRIBUTIONS

1. In experiments on CIFAR10, CIFAR100, MNIST and Diabetic Retinopathy Detection, we found that EM achieves **state-of-the-art results** when used with an appropriate calibration approach. Although BBSL and RLLS both benefit from calibration, they did not tend to outperform EM when the probabilities were well-calibrated.

2. We observed that the popular calibration approach of Temperature Scaling (TS) (Guo et al., 2017) does not tend to achieve state-of-the art results in the context of adaptation to label shift, possibly owing to large systematic biases in the calibrated probabilities (**Fig.** 1). The best results are obtained using variants of TS that contain class-specific bias capable of correcting for systematic bias.

3. We identify a theoretically-grounded strategy for computing the source-domain priors in EM-based domain adaptation that can be critically important when calibrated probability estimates have systematic bias. Empirically, we find that computing source priors in this way substantially improves the robustness of EM to the presence of systematic biases in the class probabilities. We also prove that the likelihood function of EM is concave and unimodal; thus, EM converges to the maximum likelihood estimate.

## 2 BACKGROUND

### 2.1 TEMPERATURE SCALING, VECTOR SCALING AND EXPECTED CALIBRATION ERROR

Calibration has a long history in the machine learning literature (DeGroot and Fienberg, 1983; Platt, 1999; Zadrozny and Elkan; 2002; Niculescu-Mizil and Caruana, 2005; Kuleshov and Liang, 2015; Naeini et al., 2015; Kuleshov and Ermon, 2016). In the context of modern neural networks, Guo et al. (2017) showed that Temperature Scaling, a single-parameter variant of Platt Scaling (Platt, 1999), was effective at reducing miscalibration. Temperature scaling performs calibration by introducing a temperature parameter $T$ to the logit vector of the softmax. Let $z(\boldsymbol{x}^k)$ represent a vector of the original softmax logits for example $\boldsymbol{x}^k$, and let $y_i$ be a random variable representing the label for class $i$. With temperature scaling, we have $p(y_i|\boldsymbol{x}^k) = \frac{e^{z(\boldsymbol{x}^k)_i/T}}{\sum_j e^{z(\boldsymbol{x}^k)_j/T}}$, where $T$ is optimized with respect to the Negative Log Likelihood (NLL) on a held-out portion of the training set, such as the validation set. Guo et al. (2017) compared TS to an approach defined as Vector Scaling (VS), where a different scaling parameter was used for each class along with class-specfic bias parameters. Formally, in vector scaling, $p(y_i|\boldsymbol{x}^k) = \frac{e^{(z(\boldsymbol{x}^k)_i W_i)+b_i}}{\sum_j e^{(z(\boldsymbol{x}^k)_j W_j)+b_j}}$. Guo et al. (2017) found that vector scaling had a tendency to perform slightly worse than TS as measured by a metric known as the Expected Calibration Error (Naeini et al., 2015). To compute the ECE, the predicted probabilities for the output class are partitioned into $M$ equally spaced bins, and the weighted average of the difference between

the bin's accuracy and the bin's confidence is computed, where the weights are determined by the proportion of examples falling in the bin. Formally, ECE $= \sum_{m=1}^{M} \frac{|B_m|}{n} |\text{acc}(B_m) - \text{conf}(B_m)|$, where $n$ is the number of samples.

## 2.2 LABEL SHIFT ADAPTATION VIA EXPECTATION MAXIMIZATION

In a seminal paper on label shift adaptation, Saerens et al. (2002) proposed an EM algorithm for estimating the shift in the class priors between the training and test distributions. Let $\hat{q}^{(s)}(y = i)$ denote the estimate (from EM iteration s) of the prior probability $q(y = i)$ of observing class $i$ in the test set. The algorithm proceeds as follows: first, $\hat{q}^{(0)}(y = i)$ is initialized to be equal to the class priors $\hat{p}(y = i)$ estimated from the training set. Then, the conditional probabilities in the E-step are computed as $\hat{q}^{(s)}(y = i|\boldsymbol{x}_k) = \frac{\frac{\hat{q}^{(s)}(y=i)}{\hat{p}(y=i)}\hat{p}(y=i|\boldsymbol{x}_k)}{\sum_{j=1}^{n} \frac{\hat{q}^{(s)}(y=j)}{\hat{p}(y=j)}\hat{p}(y=j|\boldsymbol{x}_k)}$.

Finally, the prior estimates in the M-step are updated as $\hat{q}^{(s+1)}(y = i) = \frac{1}{N} \sum_{k=1}^{N} \hat{q}^{(s)}(y = i|\boldsymbol{x}_k)$, where $N$ is the number of examples in the testing set. The E and M steps are iterated until convergence. As there is no need to estimate $p(\boldsymbol{x}|y)$ in any step of the EM procedure, the algorithm can scale to high-dimensional datasets. Note this procedure assumes the conditional probability estimates $\hat{p}(y = i|\boldsymbol{x}_k)$ are calibrated.

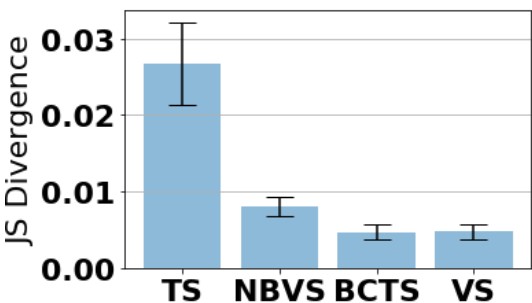

Figure 1: **Temperature Scaling exhibits systematic bias**. On CIFAR10 data, systematic bias was quantified by the Jensen-Shannon divergence between the true class label proportions and the average class predictions on a held-out test set drawn from the same distribution as the dataset used for calibration. TS: Temperature Scaling, NBVS: No-Bias Vector Scaling, BCTS: Bias-Corrected Temperature Scaling, VS: Vector Scaling. BCTS and VS had significantly lower systematic bias compared to TS and NBVS. Results are averaged over multiple models and dataset samples (**Sec. 4.1**).

## 2.3 LABEL SHIFT ADAPTATION VIA BLACK BOX SHIFT LEARNING AND REGULARIZED LEARNING UNDER LABEL SHIFTS

Following the EM approach of Saerens et al. (2002), several additional approaches for labels shift adaptation have emerged (Chan and Ng; Storkey; Schoelkopf et al., 2012; Zhang et al., 2013; Lipton et al., 2018; Azizzadenesheli et al., 2019). Many of these approaches build estimates $p(x|y)$, which can scale poorly with dataset sizes and underperform on high-dimensional data (Lipton et al., 2018). Lipton et al. (2018) proposed Black-Box Shift Learning (BBSL), which strives to efficiently estimate the weights $[\boldsymbol{w}]_i = \frac{q(y=i)}{p(y=i)}$ even in cases where the prediction model $\hat{p}(y = i|\boldsymbol{x}_k)$ is poorly calibrated or biased. BBSL proceeds as follows: let $f$ be a function that accepts an input and returns the model's predicted class, $\boldsymbol{x}_k$ denote an example from a held-out portion of the training set, and $\boldsymbol{x}'_k$ denote an example from the testing set. The empirical estimate of $\boldsymbol{w}$, denoted as $\hat{\boldsymbol{w}}$, is computed as $\hat{\boldsymbol{w}} = \hat{\boldsymbol{C}}_{\hat{y},y}^{-1}\hat{\boldsymbol{u}}_{\hat{y}}$, where $[\hat{\boldsymbol{u}}_{\hat{y}}]_i = \frac{\sum_k \mathbb{1}\{f(\boldsymbol{x}'_k)=i\}}{m}$ and $[\hat{\boldsymbol{C}}_{\hat{y},y}]_{ij} = \frac{1}{n}\sum_k \mathbb{1}\{f(\boldsymbol{x}_k) = i \text{ and } y_k = j\}$. Because the approach above is not guaranteed to produce positive values for all elements of $\hat{\boldsymbol{w}}$, any negative elements of $\hat{\boldsymbol{w}}$ are set to 0 after they are estimated. Domain adaptation is then performed by retraining the model on the entire training set distribution with examples upweighted in accordance with $\hat{\boldsymbol{w}}$. Lipton et al. (2018) denote the version of BBSL described above as **BBSL-hard**. They also compare to a variant that they call **BBSL-soft**, which they describe as the case where where $f$ outputs probabilities rather than hard classes. We interpreted this to mean $[\hat{\boldsymbol{u}}_{\hat{y}}]_i = \frac{\sum_k f(\boldsymbol{x}'_k)_i}{m}$ and $[\hat{\boldsymbol{C}}_{\hat{y},y}]_{ij} = \frac{1}{n}\sum_k f(\boldsymbol{x}_k)_i \mathbb{1}\{y_k = j\}$. Azizzadenesheli et al. (2019) further improved upon BBSL by including regularization terms in a technique known as Regularized Learning under Label Shift (RLLS). In our experiments, we compare to BBSL-hard, BBSL-soft, RLLS-hard and RLLS-soft. Regularization parameters for RLLS were set in accordance with the hard-coded values present in the publicly available code provided by the authors at `https://github.com/Angela0428/labelshift/blob/5bbe517938f4e3f5bd14c2c105de973dcc2e0917/label_shift.py#L453-L456`.

Note that BBSL and RLLS both require a portion of the training set to be held out during the initial training phase in order to accurately estimate the confusion matrix $\hat{C}_{\hat{y},y}$; in our experiments involving calibration, we use this same heldout set to calibrate the model.

## 3 METHODS

### 3.1 NO-BIAS VECTOR SCALING AND BIAS-CORRECTED TEMPERATURE SCALING

As shown in **Fig. 1**, we often found that TS alone resulted in systematically biased estimates of $p(y_i|\boldsymbol{x}^k)$, while VS, a generalization of TS that contains both class-specific bias terms and class-specific scaling terms, did not exhibit as much systematic bias. Intrigued by this observation, we investigated the performance of two intermediaries between Temperature Scaling and Vector Scaling. The first, which we refer to as No Bias Vector Scaling (NBVS), is equivalent to vector scaling but with all the class-specific bias parameters fixed at zero. The second, which we refer to as Bias-Corrected Temperature Scaling, is equivalent TS Scaling but with the addition of the class-specific bias terms from VS. As with TS and VS, the parameters are optimized to minimize the NLL on the validation set. Note that in the case of binary classification, the parameterization of BCTS reduces to Platt Scaling (Platt, 1999). Thus, BCTS can be viewed as a multi-class generalization of Platt scaling.

### 3.2 DEFINING SOURCE-DOMAIN PRIORS IN THE EM ALGORITHM

The EM algorithm of (Saerens et al., 2002) requires the user to provide estimates of the source-domain prior class probabilities $\hat{p}(y = i)$. Let us consider two possible approaches to estimating these probabilities. The first, and most obvious, is to set $\hat{p}(y = i)$ to the expected value of the binary label $y = i$ over the source domain dataset. A second, slightly less obvious approach is to set it to the expected value of $\hat{p}(y = i|x)$ over the source domain dataset, formally denoted as $\mathbf{E}_{\boldsymbol{x} \sim p(\boldsymbol{x})}[\hat{p}(y = i|\boldsymbol{x})]$. If $\hat{p}(y = i|x)$ were unbiased, we would anticipate that the two approaches agree. However, depending on the calibration of $\hat{p}(y = i|\boldsymbol{x})$, this may not be the case, bringing us to:

**Lemma 1**: In the absence of domain shift and in the limit of sufficient data, the EM algorithm will converge to the original priors $\hat{p}(y = i)$ if and only if $\hat{p}(y = i) = \mathbf{E}_{\boldsymbol{x} \sim p(\boldsymbol{x})}[\hat{p}(y = i|\boldsymbol{x})]$.

**Proof**: Note that the EM algorithm will converge when $\hat{q}^{(s+1)}(y = i) = \hat{q}^{(s)}(y = i)$. From the M-step, we know that $\hat{q}^{(s+1)}(y = i) = \frac{1}{N}\sum_{k=1}^{N}\hat{q}^{(s)}(y = i|\boldsymbol{x}_k)$, where the examples $\boldsymbol{x}_k$ are drawn from the target distribution. Substituting the formula for $\hat{q}^{(s)}(y = i|\boldsymbol{x}_k)$ from the E-step, we have

$\hat{q}^{(s+1)}(y = i) = \frac{1}{N}\sum_{k=1}^{N}\frac{\frac{\hat{q}^{(s)}(y=i)}{\hat{p}(y=i)}\hat{p}(y=i|\boldsymbol{x}_k)}{\sum_{j=1}^{n}\frac{\hat{q}^{(s)}(y=j)}{\hat{p}(y=j)}\hat{p}(y=j|\boldsymbol{x}_k)}$. To prove our lemma, we consider the scenario

where $\hat{q}(y = i) = \hat{p}(y = i)$ and check whether convergence is attained. If the samples in the target distribution are drawn from the same distribution as the source, then in the limit of sufficient $N$, the value of $\hat{q}^{(s+1)}(y = 1)$ will approach $\mathbf{E}_{x \sim p(x)}\frac{\frac{1}{1}\hat{p}(y=i|\boldsymbol{x}_k)}{\sum_{j=1}^{n}\frac{1}{1}\hat{p}(y=j|\boldsymbol{x}_k)} = \mathbf{E}_{x \sim p(x)}\hat{p}(y = i|\boldsymbol{x}_k)$. Thus, convergence at $\hat{p}(y = i)$ will be attained if and only if $\hat{p}(y = i) = \mathbf{E}_{\boldsymbol{x} \sim p(\boldsymbol{x})}[\hat{p}(y = i|\boldsymbol{x})]$ ∎

We reason that, in the absence of domain shift, it is desirable that EM converge to the original priors $\hat{p}(y = i)$. In light of Lemma 1, we set $\hat{p}(y = i)$ to be the average value of $\hat{p}(y = i|\boldsymbol{x})$ over the source-domain validation set (we use the validation set to avoid the effects of overfitting on the training set; this is the same validation set used for calibration). If we instead compute $\hat{p}(y = i)$ as the average of the binary label in the validation set, we observe very poor (even detrimental) performance with EM when the calibrated probabilities do not have bias correction (Tab. C.1).

### 3.3 METRICS FOR EVALUATING ADAPTATION TO LABEL SHIFT

The first metric we consider is the mean squared error in the true weights compared to the estimated weights (Azizzadenesheli et al., 2019). Let us denote the true target-domain prior as $q(y = i)$ and the true source domain prior as $p(y = i)$. The true class weights are defined such that $\boldsymbol{w}_i = q(y = i)/p(y = i)$. Both BBSL and RLLS directly output estimated weights $\hat{\boldsymbol{w}}_i$. For EM, the weights can be obtained by dividing the estimated target-domain priors $\hat{q}(y = i)$ by the source-domain priors $\hat{p}(y = i)$ (where the source priors are computed as described in Sec. 3.2). The mean squared error of the weights is then simply $(1/N)\sum_i(\hat{\boldsymbol{w}}_i - \boldsymbol{w}_i)^2$, where $N$ is the number of classes.

The second metric we consider is the improvement in accuracy of the domain-adapted model predictions relative to using the original model predictions. Given the ratio $\hat{q}(y = i)/\hat{p}(y = i)$, the adapted model predictions can be computed as $\hat{q}(y = i|\boldsymbol{x}_k) = \frac{\frac{\hat{q}(y=i)}{\hat{p}(y=i)}\hat{p}(y=i|\boldsymbol{x}_k)}{\sum_j \frac{\hat{q}(y=j)}{\hat{p}(y=j)}\hat{p}(y=j|\boldsymbol{x}_k)}$, similar to the E-step of EM. For EM, we use these adapted predictions to assess accuracy. In the case of BBSL, Lipton et al. (2018) recommend retraining the model to obtain adapted predictions. Due to computational constraints, we did not perform model retraining, and thus we limit the comparisons of domain-adapted accuracy only to those calibration techniques that were used in conjunction with EM.

The third metric we consider is the Jenson-Shannon divergence between the true and estimated target-domain priors. This metric is discussed further in **Sec. B**.

## 4 RESULTS

### 4.1 EXPERIMENTAL SETUP

We evaluated the efficacy of BBSL and EM coupled to different calibration approaches on MNIST, CIFAR10, CIFAR100, and a diabetic retinopathy detection dataset. For our experiments on MNIST, we used the architecture from Azizzadenesheli et al. (2019), and for our experiments on CIFAR10 and CIFAR100, we trained ten different models, each with a different random seed, using the code from Geifman and El-Yaniv (2017). For MNIST, CIFAR10 and CIFAR100, 10K examples of the training set were reserved as a held-out validation set. Dirichlet shift was simulated on the testing set by sampling with replacement in accordance with class proportions generated by a dirichlet distribution with uniform $\alpha$ values of $0.1$, $1.0$ and $10.0$ (smaller values of $\alpha$ result in more extreme label shift). Samples from the validation set were used for calibration, EM initialization and BBSL confusion matrix estimation. Accuracy and JS Divergence were reported on the label-shifted testing set, while the calibration metrics of NLL and ECE (with 15 bins) were reported on the unshifted testing set. In addition to exploring different degrees of dirichlet shift, we also investigated how the algorithms behaved when the number of samples used in the validation and testing set were varied. For example, in experiments with $n = 8000$, only 8000 samples from the validation set and 8000 samples from the shifted testing set were presented to the domain adaptation and calibration algorithms. For each model, for a given $\alpha$ and $n$, 10 trials were performed, where each trial consisted of a different sampling (without replacement) of the validation set as well as a different sampling of the dirichlet prior and the label-shifted testing set. This resulted in a total of 100 experiments (10 for each of the 10 different models). Statistical significance was calculated using a signed Wilcoxon test with a one-sided p-value threshold of $0.01$. For MNIST and CIFAR10, we also explored "tweak one" shift (Lipton et al., 2018), where the prior of the fourth class was set to a parameter $\rho$ and the remaining class priors were set to $(1 - \rho)/9$. We explored $\rho = 0.01$ and $\rho = 0.9$.

The Kaggle Diabetic Retinopathy dataset Kaggle (2015) is a collection of retinal fundus images and an associated "grade" from 0-4, where 0 indicates healthy and 1-4 indicate progressively more severe stages of retinopathy. For our experiments, we used the publicly-available pretrained model from De Fauw (2015), but it modified so as to make predictions on only one eye at a time (specifically, we supplied the mirror image of a given eye as the input for the second eye). Because test-set labels are unavailable, we separated the validation set used during the training of the model (consisting of 3514 examples) into "pseudo-validation" and "pseudo-test" sets. Specifically, for each of 100 trials, we sampled $n$ examples from the original validation set without replacement to form a pseudo-validation set, and kept the remaining examples as the pseudo-test set. Calibration was performed on the pseudo-validation set, and calibration metrics of NLL and ECE were reported on the pseudo-test set. Domain shift was then simulated by sampling from the pseudo-test set in such a way that the proportion of "healthy" labels was set to a fraction $\rho$, and the relative proportions of diseased labels was kept the same as in the source distribution. In the source distribution, $\rho = 0.73$; for the simulated domain shift, we explored $\rho = 0.5$ and $\rho = 0.9$.

### 4.2 EM WITH APPROPRIATE CALIBRATION ACHIEVES STRONG PERFORMANCE AT ESTIMATING SHIFT WEIGHTS

We compared the performance of EM, BBSL and RLLS in the presence of different types of calibration, using both MSE and JS Divergence as metrics as described in **Sec. 3.3**. Results for MSE

are in **Tables 1, 2, 3, 4, E.8 & F.3** , and results for JS Divergence are in **Tables E.6, E.7 G.3 & H.3**. Across all datasets, we observed the following general trends: first, in the absence of calibration, BBSL and RLLS tend to outperform EM, with RLLS tending to perform the best (consistent with the results in Azizzadenesheli et al. (2019)). However, as calibration improves, so does the performance of EM. In particular, the best overall performance is achieved when using the variants of temperature scaling that contain class-specific bias parameters - namely BCTS and VS - in combination with EM.

| Shift Estimator | Calibration Method | $\alpha = 0.1$ | | | $\alpha = 1.0$ | | | $\alpha = 10$ | | |
|---|---|---|---|---|---|---|---|---|---|---|
| | | $n$=2000 | $n$=4000 | $n$=8000 | $n$=2000 | $n$=4000 | $n$=8000 | $n$=2000 | $n$=4000 | $n$=8000 |
| EM | None | 0.02799; 2.9 | 0.02484; 3.38 | 0.02057; 3.41 | 0.00572; 2.62 | 0.00392; 3.0 | 0.00315; 3.28 | 0.00222; 1.7 | 0.00112; 1.88 | 0.00068; 2.5 |
| BBSL-hard | None | 0.00961; 2.46 | 0.00367; 2.2 | 0.00222; 2.0 | 0.00353; 2.24 | 0.00209; 2.38 | 0.00105; 2.19 | 0.00285; 2.9 | 0.00144; 2.64 | 0.00067; 2.21 |
| BBSL-soft | None | **0.0084; 1.31** | **0.00306; 1.23** | **0.00193; 1.38** | **0.00289; 1.41** | 0.00159; 1.05 | 0.00078; 0.97 | 0.00212; 1.48 | **0.00104; 1.54** | **0.00054; 1.39** |
| RLLS-hard | None | 0.00895; 2.19 | 0.0036; 1.99 | 0.00221; 1.88 | 0.00352; 2.3 | 0.00209; 2.25 | 0.00105; 2.26 | 0.00285; 2.7 | 0.00144; 2.56 | 0.00067; 2.35 |
| RLLS-soft | None | **0.00733; 1.14** | **0.00295; 1.2** | **0.00192; 1.33** | **0.00287; 1.43** | 0.00159; 1.32 | 0.00078; 1.3 | **0.00212; 1.22** | **0.00104; 1.38** | **0.00054; 1.55** |
| EM | TS | 0.0306; 1.44 | 0.02824; 1.62 | 0.02403; 1.66 | 0.00673; 1.27 | 0.00483; 1.53 | 0.00387; 1.7 | 0.00239; 1.42 | 0.0012; 1.38 | 0.00069; 1.42 |
| BBSL-soft | TS | **0.00852; 0.84** | **0.00309; 0.67** | **0.00197; 0.68** | **0.00291; 0.83** | 0.00158; 0.61 | 0.00079; 0.45 | **0.00211; 0.69** | **0.00105; 0.9** | **0.00055; 0.69** |
| RLLS-soft | TS | **0.00735; 0.72** | **0.00297; 0.71** | **0.00196; 0.66** | **0.00289; 0.9** | 0.00158; 0.86 | 0.00079; 0.85 | **0.00211; 0.69** | **0.00105; 0.72** | **0.00055; 0.89** |
| EM | NBVS | **0.00326; 0.53** | **0.00211; 0.69** | **0.00161; 0.82** | **0.00173; 0.36** | **0.00105; 0.68** | **0.00062; 0.84** | 0.00193; 0.86 | **0.00091; 0.8** | 0.0005; 0.92 |
| BBSL-soft | NBVS | 0.00802; 1.27 | 0.00292; 1.17 | 0.0019; 1.15 | 0.0027; 1.26 | 0.00143; 1.06 | 0.00077; 0.93 | 0.00207; 1.19 | 0.00098; 1.13 | 0.00051; 1.0 |
| RLLS-soft | NBVS | 0.00719; 1.2 | 0.00284; 1.14 | **0.00189; 1.03** | 0.00268; 1.38 | 0.00143; 1.26 | 0.00077; 1.23 | 0.00207; 0.95 | 0.00098; 1.07 | 0.00051; 1.08 |
| EM | BCTS | **0.00138; 0.09** | **0.00075; 0.26** | **0.00054; 0.42** | **0.00163; 0.36** | **0.00099; 0.48** | **0.00052; 0.6** | **0.002; 0.78** | **0.00091; 0.7** | **0.00049; 0.8** |
| BBSL-soft | BCTS | 0.00816; 1.52 | 0.00292; 1.36 | 0.00192; 1.34 | 0.00276; 1.24 | 0.00145; 1.14 | 0.00077; 1.04 | 0.0021; 1.24 | 0.00099; 1.17 | 0.00052; 1.05 |
| RLLS-soft | BCTS | 0.00717; 1.39 | 0.00283; 1.38 | 0.00189; 1.24 | 0.00274; 1.4 | 0.00145; 1.4 | 0.00077; 1.36 | 0.0021; 0.98 | 0.00099; 1.13 | 0.00052; 1.15 |
| EM | VS | **0.00182; 0.04** | **0.00077; 0.21** | **0.00052; 0.27** | **0.00161; 0.28** | **0.00097; 0.4** | **0.00054; 0.52** | **0.002; 0.8** | **0.00091; 0.66** | **0.0005; 0.8** |
| BBSL-soft | VS | 0.0081; 1.53 | 0.0029; 1.39 | 0.00189; 1.42 | 0.00274; 1.29 | 0.00144; 1.2 | 0.00078; 1.09 | 0.0021; 1.21 | 0.00098; 1.21 | 0.00052; 1.06 |
| RLLS-soft | VS | 0.00721; 1.43 | 0.00282; 1.4 | 0.00187; 1.31 | 0.00271; 1.43 | 0.00143; 1.4 | 0.00077; 1.39 | **0.0021; 0.99** | 0.00098; 1.13 | 0.00052; 1.14 |

Table 1: **CIFAR10: Comparison of EM, BBSL and RLLS (dirichlet shift).** Value before the semicolon is the average MSE in the estimated shift weights (as defined in **Sec. 3.3**). Value after the semicolon is the average rank of a method relative to the others in the group that use the same calibration. $\alpha$ represents the dirichlet shift parameter (larger $\alpha$ corresponds to less extreme shift), $n$ represents the sample size for both the validation set and the label-shifted test set. A bold value in a group is not significantly different from the best-performing method in the group, as measured by a paired Wilcoxon test at $p < 0.01$. See **Table E.2** for an equivalent table but with statistical comparisons done across all calibration methods. EM tends to outperform BBSL and RLLS when calibration techniques involving class-specific bias parameters are used.

| Shift Estimator | Calibration Method | $\rho = 0.01$ | | | $\rho = 0.9$ | | |
|---|---|---|---|---|---|---|---|
| | | $n$=2000 | $n$=4000 | $n$=8000 | $n$=2000 | $n$=4000 | $n$=8000 |
| EM | None | 0.00219; 2.28 | 0.00112; 2.2 | 0.00072; 2.04 | 0.00998; 1.72 | 0.00648; 1.47 | 0.00528; 1.7 |
| BBSL-hard | None | 0.00235; 2.89 | 0.00123; 3.03 | 0.00083; 3.37 | 0.01183; 3.03 | 0.00796; 3.12 | 0.00652; 3.48 |
| BBSL-soft | None | 0.00186; 1.67 | 0.00099; 1.63 | 0.00063; 1.53 | 0.00926; 1.69 | 0.00488; 1.44 | **0.00336; 0.84** |
| RLLS-hard | None | 0.00235; 2.15 | 0.00123; 2.19 | 0.00083; 2.39 | 0.01099; 2.21 | 0.0076; 2.54 | 0.00633; 2.78 |
| RLLS-soft | None | **0.00186; 1.01** | **0.00099; 0.95** | **0.00063; 0.67** | **0.00875; 1.35** | **0.00478; 1.43** | **0.00335; 1.2** |
| EM | TS | 0.00183; 1.04 | **0.00091; 0.92** | **0.0006; 0.78** | 0.0062; 0.65 | 0.00325; 0.52 | **0.00199; 0.42** |
| BBSL-soft | TS | 0.00178; 1.35 | 0.00093; 1.36 | 0.0006; 1.55 | 0.00914; 1.35 | 0.00515; 1.25 | 0.00384; 1.17 |
| RLLS-soft | TS | **0.00178; 0.61** | **0.00093; 0.72** | 0.0006; 0.67 | 0.00863; 1.0 | 0.00505; 1.23 | 0.00383; 1.41 |
| EM | NBVS | **0.00177; 0.7** | **0.00088; 0.62** | **0.00056; 0.44** | **0.00181; 0.08** | **0.00088; 0.1** | **0.00044; 0.0** |
| BBSL-soft | NBVS | 0.00184; 1.46 | 0.00096; 1.5 | 0.00062; 1.72 | 0.00887; 1.63 | 0.00509; 1.43 | 0.0038; 1.38 |
| RLLS-soft | NBVS | 0.00184; 0.84 | 0.00096; 0.88 | 0.00062; 0.84 | 0.0084; 1.29 | 0.00499; 1.47 | 0.00379; 1.62 |
| EM | BCTS | **0.00173; 0.82** | **0.00087; 0.72** | **0.00056; 0.48** | **0.0007; 0.0** | **0.00043; 0.0** | **0.0003; 0.0** |
| BBSL-soft | BCTS | 0.0018; 1.42 | 0.00094; 1.46 | 0.00061; 1.7 | 0.00879; 1.65 | 0.00506; 1.45 | 0.00373; 1.31 |
| RLLS-soft | BCTS | **0.0018; 0.76** | **0.00094; 0.82** | **0.00061; 0.82** | 0.00832; 1.35 | 0.00497; 1.53 | 0.00372; 1.69 |
| EM | VS | **0.00177; 0.76** | **0.00087; 0.56** | **0.00056; 0.3** | **0.00083; 0.0** | **0.00049; 0.02** | **0.00033; 0.0** |
| BBSL-soft | VS | 0.00184; 1.4 | 0.00096; 1.53 | 0.00063; 1.77 | 0.00894; 1.67 | 0.00526; 1.51 | 0.00415; 1.36 |
| RLLS-soft | VS | 0.00184; 0.84 | 0.00096; 0.91 | 0.00063; 0.93 | 0.00843; 1.33 | 0.00515; 1.47 | 0.00413; 1.64 |

Table 2: **MNIST: Comparison of EM, BBSL and RLLS ("tweak-one" shift).** Value before the semicolon is the average MSE in the estimated shift weights. Value after semicolon is the average rank of a method relative to others in the group that use the same calibration. A bold value in a group is not significantly different from the best-performing method in the group, as measured by a paired Wilcoxon test at $p < 0.01$. See **Table F.2** for an equivalent table but with statistical comparisons done across all calibration methods. EM tends to outperform BBSL and RLLS when calibration techniques involving class-specific bias parameters are used.

We also computed the improvement in accuracy achieved by EM with different calibration methods compared to an unadapted baseline (**Tables 5, 6, 7, E.1**). Across datasets, observed that either BCTS or VS tended to achieve the best accuracy. To reconcile this with the observation in Guo et al. (2017) that VS did not give the best ECE compared to TS, we calculated the Negative Log Likelihood (NLL) of different calibration methods on an unshifted test set and found that BCTS and VS tended to achieve the best NLL, even when they did not yield the best ECE (Sec. D). Empirically, we found that the NLL corresponds better with the improvement that a calibration method will give to domain adaptation (Sec. I). This is consistent with other reports stating that computing the ECE using only information about the most confidently predicted class, as was done in Guo et al. (2017), is perhaps not the best metric (Vaicenavicius et al., 2019).

| Shift Estimator | Calibration Method | α = 0.1 | | | α = 1.0 | | | α = 10.0 | | |
|---|---|---|---|---|---|---|---|---|---|---|
| | | n=7000 | n=8500 | n=10000 | n=7000 | n=8500 | n=10000 | n=7000 | n=8500 | n=10000 |
| EM | None | 2.26413; 3.01 | 2.13137; 3.18 | 2.08096; 3.22 | 0.75139; 3.52 | 0.6941; 3.62 | 0.66819; 3.69 | 0.41269; 3.75 | 0.38438; 3.86 | 0.36558; 3.94 |
| BBSL-hard | None | 1.7799; 2.94 | 1.27283; 2.88 | 1.19495; 2.95 | 0.4737; 3.04 | 0.39212; 2.95 | 0.35386; 3.04 | 0.2997; 3.08 | 0.24168; 3.03 | 0.2161; 3.03 |
| BBSL-soft | None | 1.32248; 1.73 | 0.94221; 1.79 | 0.83588; 1.79 | 0.32731; 1.62 | 0.27302; 1.72 | 0.24683; 1.68 | 0.21342; 1.64 | 0.16996; 1.6 | 0.14857; 1.59 |
| RLLS-hard | None | 0.89184; 1.56 | 0.74954; 1.51 | 0.71544; 1.41 | 0.31391; 1.47 | 0.26279; 1.38 | 0.23164; 1.32 | 0.18167; 1.3 | 0.15771; 1.34 | 0.14006; 1.31 |
| RLLS-soft | None | **0.73652; 0.76** | **0.61146; 0.64** | **0.57115; 0.63** | **0.22919; 0.35** | **0.19488; 0.33** | **0.17308; 0.27** | **0.1429; 0.23** | **0.12089; 0.17** | **0.1065; 0.13** |
| EM | TS | 0.85732; 1.0 | 0.73074; 0.99 | 0.65051; 0.99 | 0.34451; 1.32 | 0.30896; 1.54 | 0.28795; 1.41 | 0.17923; 1.4 | 0.15609; 1.52 | 0.14494; 1.68 |
| BBSL-soft | TS | 1.0511; 1.16 | 0.73046; 1.17 | 0.61651; 1.17 | 0.25082; 1.08 | 0.20128; 1.08 | 0.17385; 1.08 | 0.15657; 1.03 | 0.11901; 0.95 | 0.10114; 0.89 |
| RLLS-soft | TS | 0.70936; 0.84 | 0.58352; 0.84 | 0.52749; 0.84 | **0.20268; 0.6** | **0.1665; 0.49** | **0.14336; 0.51** | **0.12306; 0.57** | **0.09967; 0.53** | **0.08668; 0.43** |
| EM | NBVS | **0.28904; 0.49** | **0.27676; 0.48** | **0.26944; 0.55** | **0.15848; 0.63** | **0.14828; 0.72** | **0.14304; 0.94** | **0.11329; 0.75** | **0.10635; 1.09** | 0.10256; 1.3 |
| BBSL-soft | NBVS | 1.01696; 1.48 | 0.69643; 1.47 | 0.60503; 1.46 | 0.24203; 1.47 | 0.19391; 1.5 | 0.16837; 1.44 | 0.15685; 1.5 | 0.12001; 1.27 | 0.10221; 1.2 |
| RLLS-soft | NBVS | 0.65047; 1.03 | 0.52242; 1.05 | 0.48347; 0.99 | **0.19225; 0.9** | **0.15747; 0.78** | **0.13543; 0.62** | **0.12045; 0.75** | **0.09735; 0.64** | **0.08459; 0.5** |
| EM | BCTS | **0.2458; 0.33** | **0.25185; 0.38** | **0.25628; 0.4** | **0.14527; 0.5** | **0.14006; 0.65** | **0.13766; 0.84** | **0.10338; 0.63** | **0.09803; 0.94** | 0.09538; 1.2 |
| BBSL-soft | BCTS | 0.97278; 1.56 | 0.68114; 1.53 | 0.59169; 1.54 | 0.24328; 1.58 | 0.19399; 1.55 | 0.16944; 1.48 | 0.15524; 1.56 | 0.11855; 1.38 | 0.10079; 1.24 |
| RLLS-soft | BCTS | 0.63399; 1.11 | 0.51168; 1.09 | 0.47275; 1.06 | 0.19027; 0.92 | **0.15529; 0.8** | **0.1341; 0.68** | 0.11849; 0.81 | **0.09528; 0.68** | **0.08269; 0.56** |
| EM | VS | **0.1994; 0.24** | **0.2011; 0.36** | **0.20436; 0.33** | **0.13788; 0.44** | **0.1307; 0.56** | **0.12736; 0.76** | **0.10468; 0.7** | **0.09869; 1.0** | 0.09667; 1.28 |
| BBSL-soft | VS | 0.94791; 1.55 | 0.66421; 1.52 | 0.57766; 1.52 | 0.23665; 1.56 | 0.18917; 1.54 | 0.16374; 1.49 | 0.1519; 1.49 | 0.116; 1.32 | 0.09866; 1.15 |
| RLLS-soft | VS | 0.64403; 1.21 | 0.52134; 1.12 | 0.47947; 1.15 | 0.1941; 1.0 | 0.15799; 0.9 | **0.1352; 0.75** | **0.11968; 0.81** | **0.09656; 0.68** | **0.08386; 0.57** |

Table 3: **CIFAR100: Comparison of EM, BBSL and RLLS (dirichlet shift).** Value before the semicolon is the avg. MSE in the estimated shift weights. Value after the semicolon is the avg. rank of a method relative to the others in the group that use the same calibration. A bold value in a group is not significantly different from the best-performing method in the group, as measured by a paired Wilcoxon test at $p < 0.01$. See **Table G.1** for an equivalent table but with statistical comparisons done across all calibration methods. EM tends to outperform BBSL and RLLS when calibration techniques involving class-specific bias parameters are used.

| Shift Estimator | Calibration Method | ρ = 0.5 | | | ρ = 0.9 | | |
|---|---|---|---|---|---|---|---|
| | | n=500 | n=1000 | n=1500 | n=500 | n=1000 | n=1500 |
| EM | None | **1.258; 1.03** | **0.53; 0.83** | **0.389; 0.92** | 0.112; 1.96 | 0.079; 2.41 | 0.081; 2.75 |
| BBSL-hard | None | 695.531; 2.96 | 1087.163; 3.14 | 1.746; 3.1 | 370.245; 3.25 | 284.462; 3.18 | 0.743; 2.77 |
| BBSL-soft | None | 12.221; 2.17 | 1.407; 1.94 | 0.815; 1.89 | 1.171; 2.24 | 0.098; 1.86 | 0.088; 1.78 |
| RLLS-hard | None | 2.204; 2.06 | 1.398; 2.56 | 1.064; 2.6 | 0.102; 1.6 | **0.049; 1.48** | 0.054; 1.61 |
| RLLS-soft | None | 1.953; 1.78 | 0.927; 1.53 | 0.67; 1.49 | **0.067; 0.95** | **0.041; 1.07** | **0.039; 1.09** |
| EM | TS | **1.14; 0.5** | **0.465; 0.57** | **0.334; 0.52** | 0.11; 1.18 | 0.08; 1.31 | 0.079; 1.54 |
| BBSL-soft | TS | 10.72; 1.44 | 1.286; 1.39 | 0.782; 1.4 | 0.536; 1.29 | 0.089; 1.14 | 0.071; 0.88 |
| RLLS-soft | TS | 1.866; 1.06 | 0.905; 1.04 | 0.646; 1.08 | **0.069; 0.53** | **0.046; 0.55** | **0.04; 0.58** |
| EM | NBVS | **1.18; 0.61** | **0.549; 0.65** | **0.396; 0.63** | 0.168; 1.25 | 0.125; 1.35 | 0.125; 1.59 |
| BBSL-soft | NBVS | 18.236; 1.53 | 2.241; 1.47 | 1.021; 1.39 | 2.678; 1.13 | 0.109; 0.88 | **0.067; 0.77** |
| RLLS-soft | NBVS | 1.852; 0.86 | 0.879; 0.88 | 0.751; 0.98 | **0.072; 0.62** | **0.054; 0.77** | **0.046; 0.64** |
| EM | BCTS | **1.082; 0.44** | **0.426; 0.49** | **0.304; 0.46** | **0.069; 0.65** | **0.038; 0.57** | **0.036; 0.69** |
| BBSL-soft | BCTS | 61.304; 1.57 | 1.439; 1.45 | 0.887; 1.49 | 0.747; 1.29 | 0.049; 1.2 | 0.043; 1.11 |
| RLLS-soft | BCTS | 2.412; 0.99 | 0.867; 1.06 | 0.736; 1.05 | **0.066; 1.06** | 0.043; 1.23 | **0.036; 1.2** |
| EM | VS | **1.48; 0.6** | **0.503; 0.55** | **0.347; 0.5** | **0.066; 0.7** | **0.032; 0.56** | **0.029; 0.67** |
| BBSL-soft | VS | 14.874; 1.47 | 1.359; 1.44 | 0.866; 1.43 | 0.33; 1.3 | 0.049; 1.15 | 0.042; 1.15 |
| RLLS-soft | VS | 2.243; 0.93 | 0.89; 1.01 | 0.7; 1.07 | **0.065; 1.0** | 0.042; 1.29 | 0.035; 1.18 |

Table 4: **Kaggle Diabetic Retinopathy: Comparison of EM, BBSL and RLLS.** ρ represents proportion of healthy examples in shifted domain; source domain has $\rho = 0.73$. Value before semicolon is the average MSE in the estimated shift weights. Value after the semicolon is the average rank of a method relative to others in the group that use the same calibration. A bold value in a group is not significantly different from the best-performing method in the group (paired Wilcoxon test at $p < 0.01$). See **Table H.1** for an equivalent table but with statistical comparisons done across all calibration methods. EM tends to outperform BBSL and RLLS when calibration techniques involving class-specific bias parameters are used.

| Shift Estimator | Calibration Method | α = 0.1 | | | α = 1.0 | | | α = 10 | | |
|---|---|---|---|---|---|---|---|---|---|---|
| | | n=2000 | n=4000 | n=8000 | n=2000 | n=4000 | n=8000 | n=2000 | n=4000 | n=8000 |
| EM | None | 6.986; 2.77 | 6.926; 3.17 | 6.938; 3.31 | 1.968; 3.36 | 2.016; 3.44 | 2.055; 3.69 | 0.25; 2.79 | 0.217; 3.38 | 0.263; 3.42 |
| EM | TS | 7.251; 1.68 | 7.2; 2.13 | 7.217; 2.21 | 2.127; 2.83 | 2.172; 2.92 | 2.204; 3.05 | 0.243; 3.1 | 0.225; 3.34 | 0.276; 3.37 |
| EM | NBVS | **7.324; 1.63** | **7.314; 1.59** | 7.314; 1.69 | 2.5; 1.46 | 2.592; 1.47 | 2.631; 1.45 | 0.706; 1.4 | 0.788; 1.31 | 0.84; 1.3 |
| EM | BCTS | **7.328; 1.69** | **7.337; 1.42** | **7.347; 1.4** | **2.593; 0.98** | **2.664; 1.0** | 2.688; 1.09 | **0.764; 1.24** | **0.839; 0.94** | **0.884; 0.93** |
| EM | VS | 7.255; 2.23 | **7.331; 1.69** | **7.372; 1.39** | 2.548; 1.37 | 2.652; 1.17 | **2.724; 0.72** | 0.741; 1.47 | 0.838; 1.03 | 0.889; 0.98 |

Table 5: **CIFAR10: Comparison of calibration methods when using EM adaptation to dirichlet shift, with Δ%accuracy as the metric**. Unlike BBSL and RLLS, the EM algorithm does not rely on retraining to produce domain adapted probabilities. Value before the semicolon is the average change in %accuracy relative to a baseline of no adaptation. Value after the semicolon is the average rank compared to other methods in the same column. Bold values in a column are not significantly different from the best performing method in the column, as measured by a paired Wilcoxon test at $p \leq 0.01$. Calibration techniques involving class-specific bias parameters (namely BCTS and VS) tend to achieve the best performance.

| Shift Estimator | Calibration Method | $\alpha = 0.1$ | | | $\alpha = 1.0$ | | | $\alpha = 10.0$ | | |
|---|---|---|---|---|---|---|---|---|---|---|
| | | $n$=7000 | $n$=8500 | $n$=10000 | $n$=7000 | $n$=8500 | $n$=10000 | $n$=7000 | $n$=8500 | $n$=10000 |
| EM | None | 14.41; 4.0 | 14.483; 4.0 | 14.463; 4.0 | 12.25; 4.0 | 12.292; 4.0 | 12.319; 4.0 | 11.711; 4.0 | 11.819; 4.0 | 11.829; 4.0 |
| EM | TS | 26.112; 1.63 | 26.101; 1.64 | 26.048; 1.68 | 21.625; 1.82 | 21.638; 1.9 | 21.622; 1.9 | 20.721; 1.95 | 20.875; 2.2 | 20.89; 2.05 |
| EM | NBVS | 26.332; 1.6 | 26.323; 1.73 | 26.464; 1.7 | 21.588; 1.86 | 21.711; 1.91 | 21.708; 2.04 | 20.9; 1.9 | 21.059; 1.85 | 21.032; 1.93 |
| EM | BCTS | 26.485; 1.67 | 26.638; 1.47 | **26.731; 1.44** | **21.907; 1.17** | **22.004; 1.23** | 22.015; 1.24 | **21.131; 1.1** | **21.313; 1.07** | **21.297; 1.09** |
| EM | VS | **26.889; 1.1** | **26.901; 1.16** | **26.954; 1.18** | **21.94; 1.15** | **22.097; 0.96** | **22.183; 0.82** | **21.166; 1.05** | **21.408; 0.88** | **21.36; 0.93** |

Table 6: **CIFAR100: Comparison of calibration methods when using EM adaptation to dirichlet shift, with $\Delta$%accuracy as the metric**. Unlike BBSL and RLLS, the EM algorithm does not rely on retraining to produce domain adapted probabilities. Value before the semicolon is the average change in %accuracy relative to a baseline of no adaptation. Value after the semicolon is the average rank compared to other methods in the same column. Bold values in a column are not significantly different from the best performing method in the column, as measured by a paired Wilcoxon test at $p \leq 0.01$. Calibration techniques involving class-specific bias parameters (namely BCTS and VS) tend to achieve the best performance.

| Shift Estimator | Calibration Method | $\rho = 0.5$ | | | $\rho = 0.9$ | | |
|---|---|---|---|---|---|---|---|
| | | $n$=500 | $n$=1000 | $n$=1500 | $n$=500 | $n$=1000 | $n$=1500 |
| EM | None | 1.926; 3.09 | 2.076; 3.49 | 2.196; 3.64 | 1.296; 3.42 | 1.375; 3.81 | 1.477; 3.8 |
| EM | TS | 1.902; 2.96 | 2.225; 3.17 | 2.495; 3.13 | 1.626; 3.01 | 1.923; 2.88 | 1.973; 2.97 |
| EM | NBVS | 3.23; 1.69 | 3.789; 1.49 | 4.062; 1.54 | 2.074; 2.44 | 2.266; 2.24 | 2.405; 2.17 |
| EM | BCTS | **3.766; 0.88** | **4.356; 0.74** | **4.58; 0.82** | **3.548; 0.35** | **3.567; 0.36** | **3.722; 0.44** |
| EM | VS | **3.67; 1.38** | **4.278; 1.11** | **4.545; 0.87** | **3.5; 0.78** | **3.57; 0.71** | **3.746; 0.62** |

Table 7: **Kaggle Diabetic Retinopathy: Comparison of calibration methods when using EM adaptation to domain shift, with $\Delta$%accuracy as the metric.** $\rho$ represents proportion of healthy examples in shifted domain; source distribution has $\rho = 0.73$. Unlike BBSL and RLLS, the EM algorithm does not rely on retraining to produce domain adapted probabilities. Value before the semicolon is the average change in %accuracy relative to a baseline of no adaptation. Value after the semicolon is the average rank compared to other methods in the same column. Bold values in a column are not significantly different from the best performing method in the column, as measured by a paired Wilcoxon test at $p \leq 0.01$. Calibration techniques involving class-specific bias parameters (namely BCTS and VS) tend to achieve the best performance.

## 5    DISCUSSION

In this work, we explored the effect of calibration on procedures designed to perform domain adaptation to label shift. In experiments on CIFAR10, MNIST, CIFAR100 and diabetic retinopathy detection, we found the combination of EM-based domain adaptation with an appropriate calibration approach tends to outperform BBSL and RLLS. In particular, we find that the best results are achieved when the calirbation approach incorporates class-specific bias parameters that can reduce systematic bias in the class probabilities - something that is not true of the popular Temperature Scaling approach recommended by Guo et al. (2017).

In addition, we observed that when the calibrated probabilities retain systematic bias, domain adaptation to EM is sensitive to the strategy used to compute the source-domain priors. If the source-domain priors $\hat{p}(y = i)$ are not defined in a way that mirrors the systematic bias in the predicted probabilities $\hat{p}(y = i|x)$, then EM will estimate a label shift even if the target domain is identical to the source domain (**Lemma 1**) and can produce highly detrimental results (**Tables C.1 & C.2**). By contrast, if the source domain priors for EM are initialized as recommend in **Sec. 3.2**, EM becomes substantially more tolerant of systematic bias in the calibrated probabilities, although it does not tend to outperform BBSL or RLLS in the presence of poor calibration.

One concern when using EM is the possibility of getting trapped in local minima. To address this concern, we analyzed the optimization of the likelihood function of EM and determined that it is concave **Sec. A**. Thus, we can expect that EM will converge to the global maximum of the likelihood. Future work could extend this analysis to derive generalization guarantees for the domain adaptation. In the meantime, our recommendation based on our empirical results is as follows: if there is insufficient data to calibrate the predicted probabilities, use RLLS for domain adaptation. Otherwise, use a calibration approach that incorporates class-specific bias parameters (either bias-corrected temperature scaling or vector scaling), and use Expectation Maximization for domain adaptation.

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

## A    LIKELIHOOD FUNCTION OF EM OBJECTIVE

In this section, we show that, in our label shift setting, the likelihood function is unimodal and concave. Thus, the EM approach converges to the maximum likelihood estimation.

**Lemma A**: the EM objective is concave.

**Proof**: Let $q(w_i)$ and $p(w_i)$ denote the target and source domain prior probabilities for class $i$. We wish to find target-domain priors $q(\boldsymbol{w})$ that maximize the log-likelihood function given by

$$l(\boldsymbol{X}; q(\boldsymbol{w})) = \sum_k \log \sum_i q(\boldsymbol{x}_k|w_i)q(w_i) \tag{1}$$

$$= \sum_k \log \sum_i p(\boldsymbol{x}_k|w_i)q(w_i) \tag{2}$$

$$= \sum_k \log \sum_i \frac{p(w_i|\boldsymbol{x}_k)p(\boldsymbol{x}_k)}{p(w_i)}q(w_i) \tag{3}$$

$$= \sum_k \log \left( p(\boldsymbol{x}_k) \sum_i \frac{p(w_i|\boldsymbol{x}_k)}{p(w_i)}q(w_i) \right) \tag{4}$$

$$= \sum_k \log(p(\boldsymbol{x}_k)) + \log \sum_i \frac{p(w_i|\boldsymbol{x}_k)}{p(w_i)}q(w_i) \tag{5}$$

where (2) follows from the label shift assumptions and (3) follows from Bayes' rule. Now note that the maximization is independent of $p(\boldsymbol{x}_k)$ so the optimization problem is equivalently written as follows

$$
\begin{aligned}
\max_{q(\boldsymbol{w})} \quad & \sum_k \log \sum_i \frac{p(w_i|\boldsymbol{x}_k)}{p(w_i)}q(w_i) \\
\text{s.t.} \quad & \mathbf{1}^T \cdot q(\boldsymbol{w}) = 1 \\
& q(w_i) \geq 0 \quad \forall i
\end{aligned}
\tag{6}
$$

Note that in the objective, both $p(w_i|\boldsymbol{x}_k)$ and $p(w_i)$ are constants with respect to $q(\boldsymbol{w})$. Hence, the objective function is the sum of logs of linear functions in our decision variable and the constraints are affine. Therefore, the maximization problem is concave. ∎

## B    JENSON-SHANNON DIVERGENCE METRIC FOR EVALUATION ADAPTATION TO LABEL SHIFT

In this section, we describe how we use the Jenson-Shannon divergence between the true target-domain priors and the estimated target-domain priors to evaluate a given domain adaptation method. Let us denote the true target-domain prior as $q(y = i)$ and the estimate as $q'(y = i)$. In the case of BBSL and RLLS, $q'(y = i)$ can be calculated using the class weights $\hat{\boldsymbol{w}}_i$, which are intended to estimate $q(y = i)/p(y = i)$. Specifically, we use the formula $q'(y = i) = \frac{\hat{\boldsymbol{w}}_i p'(y=i)}{\sum_j \hat{\boldsymbol{w}}_j p'(y=j)}$, where $p'(y = i)$ is defined as the average of the true class labels in the source domain.

In the case of EM, some nuance is required. In **Sec. 3.2**, we noted that when performing EM, the source domain priors $\hat{p}(y = i)$ should be defined as the average of the predictions $\hat{p}(y = i|\boldsymbol{x})$ over the examples from the source domain (rather than being defined as the average of the labels). Once the EM algorithm has reached convergence, it will output an estimate $\hat{q}(y = i)$ of the target domain priors. Naively, it would seem that we could use $\hat{q}(y = i)$ as our estimate of $q(y = i)$ when computing the Jensen-Shannon divergence. However, if $\hat{p}(y = i)$ is systematically biased, the target domain priors output by EM would likely also be systematically biased. We found that we can obtain superior estimates of $q(y = i)$ by leveraging the true labels in the source domain as follows: first, we

average the labels in the source domain to obtain to obtain $p'(y = i)$, which is an unbiased estimate of $p(y = i)$. Then, we use the ratio $\hat{q}(y = i)/\hat{p}(y = i)$ from EM as a shift estimate. That is, we compute our estimate of the target domain priors using the formula $q'(y = i) = \frac{\frac{\hat{q}(y=i)}{\hat{p}(y=i)} p'(y=i)}{\sum_j \frac{\hat{q}(y=j)}{\hat{p}(y=j)} p'(y=j)}$.

We find that target domain priors computed in this way can occasionally perform surprisingly well even when the predictions themselves are not very well calibrated.

## C  COMPARISON OF STRATEGIES FOR INITIALIZING EM SOURCE PROBABILITIES

| Shift Estimator | Calibration Method | $\rho = 0.5$ | | | $\rho = 0.9$ | | |
|---|---|---|---|---|---|---|---|
| | | $n$=500 | $n$=1000 | $n$=1500 | $n$=500 | $n$=1000 | $n$=1500 |
| EM: source priors from preds | None | **1.926; 0.0** | **2.076; 0.0** | **2.196; 0.0** | **1.296; 0.26** | **1.375; 0.17** | **1.477; 0.14** |
| EM: source priors from labels | None | -3.488; 1.0 | -3.541; 1.0 | -3.382; 1.0 | 0.782; 0.74 | 0.937; 0.83 | 1.043; 0.86 |
| EM: source priors from preds | TS | **1.902; 0.0** | **2.225; 0.0** | **2.495; 0.0** | **1.626; 0.0** | **1.923; 0.0** | **1.973; 0.0** |
| EM: source priors from labels | TS | -56.162; 1.0 | -62.552; 1.0 | -64.195; 1.0 | -69.146; 1.0 | -76.619; 1.0 | -83.083; 1.0 |
| EM: source priors from preds | NBVS | **3.23; 0.0** | **3.789; 0.0** | **4.062; 0.0** | **2.074; 0.02** | **2.266; 0.01** | **2.405; 0.02** |
| EM: source priors from labels | NBVS | -9.448; 1.0 | -5.134; 1.0 | -4.772; 1.0 | -2.616; 0.98 | 0.431; 0.99 | 0.631; 0.98 |
| EM: source priors from preds | BCTS | **3.766; 0.0** | 4.356; 0.03 | 4.58; 0.01 | 3.548; 0.0 | 3.567; 0.01 | 3.722; 0.01 |
| EM: source priors from labels | BCTS | 3.764; 1.0 | **4.357; 0.97** | 4.58; 0.99 | 3.548; 1.0 | **3.568; 0.99** | **3.723; 0.99** |
| EM: source priors from preds | VS | 3.67; 0.08 | 4.278; 0.08 | 4.545; 0.08 | 3.5; 0.03 | 3.57; 0.03 | 3.746; 0.03 |
| EM: source priors from labels | VS | 3.662; 0.92 | 4.278; 0.92 | **4.559; 0.92** | **3.506; 0.97** | **3.572; 0.97** | 3.746; 0.97 |

Table C.1: **The strategy for computing EM source priors heavily affects domain adaptation if probabilities retain systematic bias**. Value before the semicolon is the average improvement in %accuracy (across 100 trials) caused by applying domain adaptation to the predictions on a diabetic retinopathy prediction task. Value after the semicolon is the average rank of a particular method relative to the other method in the pair. Domain shift is induced by varying the proportion of "healthy" examples $\rho$; in the source distribution, $\rho = 0.73$. We see that calibration methods that lack class-specific bias parameters (i.e. no calibration, TS and NBVS) can hurt domain adaptation if source priors are initialized by averaging true labels rather than the predicted probabilities. A bold value in a pair is significantly better than the non-bold value according to a paired Wilcoxon test at $p \leq 0.01$. See **Table C.2** for analogous results using JS Div. See **Sec. 4.1** for details on the experimental setup.

| Shift Estimator | Calibration Method | $\rho = 0.5$ | | | $\rho = 0.9$ | | |
|---|---|---|---|---|---|---|---|
| | | $n$=500 | $n$=1000 | $n$=1500 | $n$=500 | $n$=1000 | $n$=1500 |
| EM: source priors from preds | None | **0.077; 0.0** | **0.059; 0.0** | **0.054; 0.0** | **0.111; 0.4** | **0.1; 0.37** | **0.102; 0.32** |
| EM: source priors from labels | None | 0.253; 1.0 | 0.256; 1.0 | 0.256; 1.0 | 0.125; 0.6 | 0.116; 0.63 | 0.114; 0.68 |
| EM: source priors from preds | TS | **0.09; 0.0** | **0.068; 0.0** | **0.061; 0.0** | **0.104; 0.0** | **0.094; 0.0** | **0.094; 0.0** |
| EM: source priors from labels | TS | 0.61; 1.0 | 0.628; 1.0 | 0.647; 1.0 | 0.629; 1.0 | 0.639; 1.0 | 0.643; 1.0 |
| EM: source priors from preds | NBVS | **0.107; 0.0** | **0.089; 0.0** | **0.079; 0.0** | **0.11; 0.0** | **0.1; 0.0** | **0.102; 0.0** |
| EM: source priors from labels | NBVS | 0.348; 1.0 | 0.327; 1.0 | 0.32; 1.0 | 0.214; 1.0 | 0.192; 1.0 | 0.188; 1.0 |
| EM: source priors from preds | BCTS | 0.111; 0.4 | 0.095; 0.44 | 0.082; 0.41 | 0.078; 0.49 | 0.065; 0.54 | 0.063; 0.52 |
| EM: source priors from labels | BCTS | 0.111; 0.6 | 0.095; 0.56 | 0.082; 0.59 | 0.078; 0.51 | 0.065; 0.46 | 0.063; 0.48 |
| EM: source priors from preds | VS | 0.108; 0.43 | 0.088; 0.5 | 0.076; 0.48 | 0.077; 0.58 | 0.062; 0.45 | 0.059; 0.54 |
| EM: source priors from labels | VS | 0.108; 0.57 | 0.088; 0.5 | 0.076; 0.52 | 0.077; 0.42 | 0.062; 0.55 | 0.059; 0.46 |

Table C.2: Similar to **Table C.1**, but using Jensen-Shannon Divergence as a metric to assess the quality of domain adaptation. See **Sec. B** for a description of how the Jensen-Shannon Divergence metric is calculated.

## D  CALIBRATION QUALITY COMPARISON

We find that bias-corrected versions of Temperature Scaling (namely Bias-Corrected Temperature Scaling and Vector Scaling) tend to yield the best Negative Log Likelihood on an unshifted test set, even if they do not always yield the best ECE. Results are shown in the tables below.

| Calibration | NLL | | | ECE | | |
|---|---|---|---|---|---|---|
| Method | $n$=2000 | $n$=4000 | $n$=8000 | $n$=2000 | $n$=4000 | $n$=8000 |
| None | 0.299; 4.0 | 0.299; 4.0 | 0.299; 4.0 | 2.726; 4.0 | 2.726; 4.0 | 2.726; 4.0 |
| TS | 0.291; 2.99 | 0.291; 3.0 | 0.291; 3.0 | **1.069; 1.23** | 1.06; 1.83 | 1.027; 2.12 |
| NBVS | 0.277; 1.67 | 0.275; 1.92 | 0.274; 1.99 | 1.109; 1.51 | 1.023; 1.51 | **0.952; 1.35** |
| BCTS | **0.274; 0.34** | **0.272; 0.54** | 0.271; 0.71 | **1.06; 1.02** | 0.987; 1.02 | 0.937; 1.06 |
| VS | 0.275; 1.0 | **0.272; 0.54** | **0.271; 0.3** | 1.161; 2.24 | 1.035; 1.64 | 0.976; 1.47 |

Table D.1: **CIFAR10: NLL and ECE for different calibration methods**. Metrics were computed on a test set that had the same distribution as the validation set. Value before the semicolon is the average of the metric over all the runs. Value after the semicolon is the average rank of the method relative to other methods in the column. $n$ indicates the number of examples used for calibratin. Bold values in a column are not significantly different from the best performing method in the column, as measured by a paired Wilcoxon test at $p \leq 0.01$. See **Sec. 4.1** for details on the experimental setup.

| Calibration | NLL | | | ECE | | |
|---|---|---|---|---|---|---|
| Method | $n$=7000 | $n$=8500 | $n$=10000 | $n$=7000 | $n$=8500 | $n$=10000 |
| None | 1.735; 4.0 | 1.735; 4.0 | 1.735; 4.0 | 20.041; 4.0 | 20.041; 4.0 | 20.041; 4.0 |
| TS | 1.286; 3.0 | 1.286; 3.0 | 1.286; 3.0 | 3.134; 2.87 | 3.151; 2.87 | 3.135; 2.9 |
| NBVS | 1.241; 2.0 | 1.24; 2.0 | 1.239; 2.0 | **2.263; 0.09** | **2.281; 0.1** | **2.324; 0.1** |
| BCTS | 1.234; 0.71 | 1.233; 0.9 | 1.232; 1.0 | 2.879; 2.11 | 2.9; 2.12 | 2.881; 2.1 |
| VS | **1.234; 0.29** | **1.231; 0.1** | **1.229; 0.0** | 2.458; 0.93 | 2.48; 0.91 | 2.456; 0.9 |

Table D.2: **CIFAR100: NLL and ECE for different calibration methods**. Analogous to **Table D.1**.

| Calibration | NLL | | | ECE | | |
|---|---|---|---|---|---|---|
| Method | $n$=500 | $n$=1000 | $n$=1500 | $n$=500 | $n$=1000 | $n$=1500 |
| None | 0.64; 4.0 | 0.639; 4.0 | 0.639; 4.0 | 8.734; 4.0 | 8.737; 4.0 | 8.767; 4.0 |
| TS | 0.571; 3.0 | 0.57; 3.0 | 0.569; 3.0 | 3.65; 2.77 | 3.729; 2.92 | 3.853; 2.76 |
| NBVS | 0.543; 2.0 | 0.54; 2.0 | 0.539; 2.0 | **2.13; 0.67** | 2.028; 0.97 | 2.129; 1.01 |
| BCTS | **0.514; 0.21** | **0.511; 0.57** | 0.511; 0.63 | 2.255; 1.21 | **2.097; 1.17** | **2.171; 1.14** |
| VS | 0.518; 0.79 | **0.512; 0.43** | **0.51; 0.37** | 2.323; 1.35 | **2.065; 0.94** | **2.153; 1.09** |

Table D.3: **Kaggle Diabetic Retinopathy Detection: NLL and ECE for different calibration methods**. Analogous to **Table D.1**.

# E CIFAR10 SUPPLEMENTARY TABLES

| Shift Estimator | Calibration Method | $\rho = 0.01$ | | | $\rho = 0.9$ | | |
|---|---|---|---|---|---|---|---|
| | | n=2000 | n=4000 | n=8000 | n=2000 | n=4000 | n=8000 |
| EM | None | 0.784; 2.91 | 0.798; 3.04 | 0.761; 3.31 | 16.304; 3.79 | 16.356; 3.84 | 16.369; 3.92 |
| EM | TS | 0.807; 2.92 | 0.807; 3.14 | 0.775; 3.4 | 17.193; 2.48 | 17.26; 2.67 | 17.288; 2.75 |
| EM | NBVS | **1.149; 1.31** | 1.172; 1.56 | 1.199; 1.39 | 17.588; 1.52 | 17.674; 1.51 | 17.738; 1.68 |
| EM | BCTS | **1.175; 1.38** | 1.224; 1.27 | 1.262; 1.15 | **17.724; 1.09** | 17.779; 1.17 | 17.84; 1.24 |
| EM | VS | **1.182; 1.48** | **1.258; 0.99** | **1.301; 0.75** | **17.727; 1.12** | **17.874; 0.81** | **17.988; 0.41** |

Table E.1: **CIFAR10: Comparison of calibration methods when using EM adaptation to "tweak-one" shift, with $\Delta\%$ accuracy as the metric.** Analogous to **Table 5**.

| Shift Estimator | Calibration Method | $\alpha = 0.1$ | | | $\alpha = 1.0$ | | | $\alpha = 10$ | | |
|---|---|---|---|---|---|---|---|---|---|---|
| | | n=2000 | n=4000 | n=8000 | n=2000 | n=4000 | n=8000 | n=2000 | n=4000 | n=8000 |
| EM | None | 0.02799; 12.09 | 0.02484; 13.62 | 0.02057; 13.97 | 0.00572; 11.84 | 0.00392; 12.83 | 0.00315; 13.69 | 0.00222; 9.11 | 0.00112; 9.44 | 0.00068; 11.34 |
| EM | TS | 0.0306; 11.82 | 0.02824; 12.83 | 0.02403; 12.94 | 0.00673; 10.83 | 0.00483; 12.24 | 0.00387; 12.94 | 0.00239; 10.53 | 0.0012; 10.74 | 0.00069; 11.23 |
| EM | NBVS | 0.00326; 4.77 | 0.00211; 6.06 | 0.00161; 6.48 | 0.00173; 3.85 | 0.00105; 5.15 | 0.00062; 5.74 | **0.00193; 6.28** | **0.00091; 5.7** | **0.0005; 6.61** |
| EM | BCTS | **0.00138; 1.37** | **0.00075; 2.32** | **0.00054; 3.04** | **0.00163; 2.69** | **0.00099; 3.79** | **0.00052; 4.26** | **0.002; 6.19** | **0.00091; 5.52** | **0.00049; 5.99** |
| EM | VS | **0.00182; 1.35** | **0.00077; 1.85** | **0.00052; 2.12** | **0.00161; 2.73** | **0.00097; 3.66** | **0.00054; 4.04** | **0.002; 6.35** | **0.00091; 5.82** | **0.0005; 6.69** |
| BBSL-hard | None | 0.00961; 11.15 | 0.00367; 10.57 | 0.00222; 9.94 | 0.00353; 11.03 | 0.00209; 10.78 | 0.00105; 10.77 | 0.00285; 11.97 | 0.00144; 11.23 | 0.00067; 9.99 |
| BBSL-soft | None | 0.0084; 8.77 | 0.00306; 8.34 | 0.00193; 8.24 | 0.00289; 8.89 | 0.00159; 8.42 | 0.00078; 7.41 | 0.00212; 7.85 | 0.00104; 8.22 | 0.00054; 7.7 |
| BBSL-soft | TS | 0.00852; 8.29 | 0.00309; 7.73 | 0.00197; 7.9 | 0.00291; 8.65 | 0.00158; 7.49 | 0.00079; 7.54 | 0.00211; 7.44 | 0.00105; 8.36 | 0.00055; 8.01 |
| BBSL-soft | NBVS | 0.00802; 8.83 | 0.00292; 8.2 | 0.0019; 8.1 | 0.0027; 7.92 | 0.00143; 7.0 | 0.00077; 7.21 | **0.00207; 7.35** | 0.00098; 6.83 | **0.00051; 6.69** |
| BBSL-soft | BCTS | 0.00816; 8.49 | 0.00292; 7.68 | 0.00192; 7.72 | 0.00276; 8.02 | 0.00145; 7.69 | 0.00077; 6.58 | 0.0021; 7.33 | 0.00099; 7.53 | 0.00052; 6.77 |
| BBSL-soft | VS | 0.0081; 8.54 | 0.0029; 8.51 | 0.00189; 8.01 | 0.00274; 7.65 | 0.00144; 7.36 | 0.00078; 7.35 | 0.0021; 7.52 | 0.00098; 7.52 | 0.00052; 7.53 |
| RLLS-hard | None | 0.00895; 10.59 | 0.0036; 10.15 | 0.00221; 9.77 | 0.00352; 11.08 | 0.00209; 10.63 | 0.00105; 10.83 | 0.00285; 11.77 | 0.00144; 11.15 | 0.00067; 10.13 |
| RLLS-soft | None | 0.00733; 8.23 | 0.00295; 8.03 | 0.00192; 7.92 | 0.00287; 8.81 | 0.00159; 8.7 | 0.00078; 7.75 | 0.00212; 7.59 | 0.00104; 8.06 | 0.00054; 7.86 |
| RLLS-soft | TS | 0.00735; 7.75 | 0.00297; 7.39 | 0.00196; 7.63 | 0.00289; 8.57 | 0.00158; 7.69 | 0.00079; 7.93 | 0.00211; 7.24 | 0.00105; 8.18 | 0.00055; 8.21 |
| RLLS-soft | NBVS | 0.00719; 8.33 | 0.00284; 7.62 | 0.00189; 7.59 | 0.00268; 7.83 | 0.00143; 7.16 | 0.00077; 7.46 | **0.00207; 7.11** | 0.00098; 6.77 | **0.00051; 6.77** |
| RLLS-soft | BCTS | 0.00717; 7.78 | 0.00283; 7.14 | 0.00189; 7.09 | 0.00274; 7.99 | 0.00145; 7.88 | 0.00077; 6.87 | 0.0021; 7.07 | 0.00099; 7.49 | 0.00052; 6.87 |
| RLLS-soft | VS | 0.00721; 7.85 | 0.00282; 7.96 | 0.00187; 7.54 | 0.00271; 7.62 | 0.00143; 7.53 | 0.00077; 7.63 | 0.0021; 7.3 | 0.00098; 7.44 | 0.00052; 7.61 |

Table E.2: **CIFAR10: Comparison of all calibration and domain adaptation methods, using MSE (Sec. 3.3) as the metric (dirichlet shift).** Value before the semicolon is the average of the metric over all trials. Value after the semicolon is the average rank of the domain adaptation + calibration method combination relative to the other method combinations in the column. Bold values in a column are not significantly different from the best-performing method in the column as measured by a paired Wilcoxon test at $p < 0.01$. EM with BCTS or VS tends to achieve the best performance. See **Sec. 4.1** for details on the experimental setup.

| Shift Estimator | Calibration Method | $\rho = 0.01$ | | | $\rho = 0.9$ | | |
|---|---|---|---|---|---|---|---|
| | | n=2000 | n=4000 | n=8000 | n=2000 | n=4000 | n=8000 |
| EM | None | 0.00202; 9.59 | 0.00104; 11.31 | 0.00069; 13.25 | 0.08233; 11.91 | 0.07773; 13.38 | 0.0756; 14.63 |
| EM | TS | 0.00212; 10.33 | 0.00111; 11.69 | 0.00074; 13.8 | 0.11247; 11.8 | 0.10919; 12.63 | 0.10942; 13.26 |
| EM | NBVS | **0.0015; 5.05** | **0.00066; 5.01** | 0.00041; 6.2 | 0.0041; 2.84 | 0.00263; 3.75 | 0.00206; 4.8 |
| EM | BCTS | **0.0015; 4.71** | **0.00065; 4.52** | **0.0004; 4.89** | **0.00221; 1.38** | **0.00137; 1.64** | **0.00104; 1.72** |
| EM | VS | **0.00151; 5.09** | **0.00064; 4.14** | **0.0004; 5.04** | **0.00266; 1.08** | **0.00144; 1.19** | **0.00104; 1.25** |
| BBSL-hard | None | 0.00258; 12.4 | 0.00119; 12.01 | 0.00059; 10.53 | 0.02157; 10.62 | 0.01198; 10.15 | 0.00599; 8.98 |
| BBSL-soft | None | 0.00183; 8.21 | 0.00081; 8.03 | 0.00045; 6.85 | 0.01905; 9.79 | 0.01102; 9.49 | 0.00612; 9.55 |
| BBSL-soft | TS | 0.00181; 7.56 | 0.0008; 7.34 | 0.00044; 6.67 | 0.0201; 9.89 | 0.01171; 10.23 | 0.00677; 10.92 |
| BBSL-soft | NBVS | 0.00175; 8.07 | 0.0008; 7.62 | 0.00047; 7.67 | 0.01791; 7.96 | 0.00921; 7.03 | 0.00498; 6.99 |
| BBSL-soft | BCTS | 0.00174; 7.39 | 0.00081; 7.74 | 0.00046; 7.07 | 0.01808; 7.96 | 0.00937; 7.28 | 0.00501; 6.17 |
| BBSL-soft | VS | 0.00176; 8.27 | 0.00081; 7.91 | 0.00047; 8.73 | 0.01798; 7.74 | 0.00912; 6.35 | 0.00486; 5.51 |
| RLLS-hard | None | 0.00256; 12.13 | 0.00119; 11.7 | 0.00059; 10.11 | 0.02055; 10.47 | 0.01172; 10.43 | 0.00596; 9.35 |
| RLLS-soft | None | 0.00182; 7.8 | 0.00081; 7.77 | 0.00045; 6.51 | 0.01872; 9.44 | 0.011; 9.9 | 0.0061; 10.24 |
| RLLS-soft | TS | 0.00179; 7.24 | 0.0008; 7.06 | 0.00044; 6.37 | 0.0198; 10.82 | 0.0117; 10.82 | 0.00675; 11.68 |
| RLLS-soft | NBVS | 0.00174; 7.55 | 0.0008; 7.24 | 0.00047; 7.29 | 0.01757; 7.92 | 0.00916; 7.54 | 0.00496; 7.73 |
| RLLS-soft | BCTS | 0.00173; 6.9 | 0.00081; 7.38 | 0.00046; 6.71 | 0.01769; 7.76 | 0.0093; 7.58 | 0.00499; 6.96 |
| RLLS-soft | VS | 0.00175; 7.71 | 0.00081; 7.53 | 0.00047; 8.31 | 0.01759; 7.64 | 0.00906; 6.61 | 0.00484; 6.26 |

Table E.3: **CIFAR10: Comparison of all calibration and domain adaptation methods, using MSE (Sec. 3.3) as the metric ("tweak-one" shift).** Value before the semicolon is the average of the metric over all trials. Value after the semicolon is the average rank of the domain adaptation + calibration method combination relative to the other method combinations in the column. Bold values in a column are not significantly different from the best-performing method in the column as measured by a paired Wilcoxon test at $p < 0.01$. EM with BCTS or VS tends to achieve the best performance. See **Sec. 4.1** for details on the experimental setup.

| Shift Estimator | Calibration Method | $\alpha = 0.1$ | | | $\alpha = 1.0$ | | | $\alpha = 10$ | | |
|---|---|---|---|---|---|---|---|---|---|---|
| | | n=2000 | n=4000 | n=8000 | n=2000 | n=4000 | n=8000 | n=2000 | n=4000 | n=8000 |
| EM | None | 0.035; 4.72 | 0.033; 5.08 | 0.033; 6.45 | 0.021; 7.32 | 0.018; 9.25 | 0.016; 11.23 | 0.016; 7.84 | 0.011; 8.76 | 0.009; 9.93 |
| EM | TS | 0.024; 2.14 | 0.022; 2.8 | 0.021; 3.31 | 0.019; 5.65 | 0.017; 8.67 | 0.01; 9.55 | 0.016; 8.75 | 0.012; 9.73 | 0.009; 10.19 |
| EM | NBVS | **0.021; 1.56** | **0.017; 1.72** | 0.017; 1.9 | 0.017; 3.49 | **0.013; 3.68** | **0.01; 3.99** | **0.015; 6.24** | **0.01; 5.97** | **0.008; 6.74** |
| EM | BCTS | **0.02; 1.38** | **0.016; 1.25** | **0.015; 1.38** | **0.016; 2.61** | **0.012; 3.03** | **0.009; 2.87** | **0.015; 6.24** | **0.01; 5.74** | **0.008; 6.18** |
| EM | VS | 0.023; 2.12 | **0.017; 1.56** | **0.015; 1.5** | **0.016; 3.07** | **0.013; 2.95** | **0.009; 3.74** | **0.015; 6.59** | **0.01; 6.06** | **0.008; 6.96** |
| BBSL-hard | None | 0.076; 13.38 | 0.064; 13.39 | 0.055; 13.16 | 0.03; 11.94 | 0.023; 12.03 | 0.016; 11.91 | 0.019; 12.06 | 0.013; 11.07 | 0.009; 10.1 |
| BBSL-soft | None | 0.068; 11.41 | 0.056; 11.37 | 0.049; 11.28 | 0.027; 9.94 | 0.019; 9.02 | 0.014; 8.36 | 0.016; 8.53 | 0.011; 8.52 | 0.008; 7.81 |
| BBSL-soft | TS | 0.067; 10.36 | 0.056; 10.58 | 0.048; 10.29 | 0.026; 9.08 | 0.019; 8.36 | 0.013; 7.73 | 0.016; 7.94 | 0.011; 8.45 | 0.008; 8.28 |
| BBSL-soft | NBVS | 0.067; 10.79 | 0.055; 10.51 | 0.048; 10.47 | 0.026; 8.98 | 0.018; 7.89 | 0.013; 7.89 | **0.016; 7.27** | 0.011; 7.12 | **0.008; 6.83** |
| BBSL-soft | BCTS | 0.066; 10.09 | 0.055; 10.03 | 0.047; 9.52 | 0.025; 8.52 | 0.018; 8.07 | 0.014; 8.02 | 0.016; 7.44 | 0.011; 7.47 | **0.008; 6.91** |
| BBSL-soft | VS | 0.066; 10.23 | 0.055; 10.41 | 0.047; 9.67 | 0.025; 8.51 | 0.018; 8.4 | 0.014; 8.2 | 0.016; 7.53 | 0.011; 7.38 | 0.008; 7.58 |
| RLLS-hard | None | 0.074; 12.48 | 0.063; 12.24 | 0.054; 12.17 | 0.03; 11.95 | 0.023; 11.98 | 0.016; 11.91 | 0.019; 11.86 | 0.013; 11.13 | 0.009; 10.3 |
| RLLS-soft | None | 0.066; 10.01 | 0.055; 9.94 | 0.048; 9.95 | 0.027; 9.95 | 0.019; 9.14 | 0.014; 8.46 | 0.016; 8.33 | 0.011; 8.46 | 0.008; 8.03 |
| RLLS-soft | TS | 0.066; 8.94 | 0.055; 9.15 | 0.048; 9.13 | 0.026; 9.02 | 0.019; 8.54 | 0.013; 7.87 | 0.016; 7.76 | 0.011; 8.37 | 0.008; 8.48 |
| RLLS-soft | NBVS | 0.066; 9.2 | 0.054; 8.93 | 0.047; 8.99 | 0.026; 9.01 | 0.018; 8.09 | 0.013; 7.96 | **0.016; 7.09** | 0.011; 7.02 | **0.008; 6.93** |
| RLLS-soft | BCTS | 0.065; 8.46 | 0.054; 8.39 | 0.046; 8.34 | 0.025; 8.48 | 0.018; 8.3 | 0.014; 8.07 | 0.016; 7.24 | 0.011; 7.43 | **0.008; 7.03** |
| RLLS-soft | VS | 0.065; 8.73 | 0.054; 8.65 | 0.046; 8.49 | 0.026; 8.48 | 0.018; 8.6 | 0.014; 8.24 | 0.016; 7.29 | 0.011; 7.32 | 0.008; 7.72 |

Table E.4: **CIFAR10: Comparison of all calibration and domain adaptation methods, using JS Divergence (Sec. B) as the metric (dirichlet shift).** Value before the semicolon is the average of the metric over all trials. Value after the semicolon is the average rank of the domain adaptation + calibration method combination relative to the other method combinations in the column. Bold values in a column are not significantly different from the best-performing method in the column as measured by a paired Wilcoxon test at $p < 0.01$. EM with BCTS or VS tends to achieve the best performance. See **Sec. 4.1** for details on the experimental setup.

| Shift Estimator | Calibration Method | $\rho = 0.01$ | | | $\rho = 0.9$ | | |
|---|---|---|---|---|---|---|---|
| | | n=2000 | n=4000 | n=8000 | n=2000 | n=4000 | n=8000 |
| EM | None | 0.018; 7.31 | 0.013; 8.12 | 0.012; 11.66 | 0.047; 4.59 | 0.045; 6.84 | 0.044; 10.62 |
| EM | TS | 0.02; 8.76 | 0.015; 9.26 | 0.014; 11.54 | 0.036; 2.51 | 0.034; 3.4 | 0.032; 5.7 |
| EM | NBVS | **0.016; 4.61** | **0.01; 4.73** | **0.008; 5.66** | 0.033; 2.1 | 0.028; 2.65 | 0.026; 3.77 |
| EM | BCTS | **0.016; 5.29** | **0.01; 4.73** | 0.009; 5.77 | **0.028; 1.0** | **0.023; 1.35** | **0.021; 1.79** |
| EM | VS | **0.016; 5.07** | **0.01; 4.02** | **0.008; 4.25** | 0.029; 1.32 | 0.023; 1.14 | **0.02; 0.88** |
| BBSL-hard | None | 0.026; 11.55 | 0.018; 11.66 | 0.013; 11.01 | 0.096; 14.24 | 0.078; 13.7 | 0.063; 13.76 |
| BBSL-soft | None | 0.021; 9.03 | 0.014; 8.29 | 0.01; 7.93 | 0.08; 11.66 | 0.064; 11.64 | 0.045; 11.17 |
| BBSL-soft | TS | 0.021; 8.38 | 0.014; 7.45 | 0.01; 7.39 | 0.078; 10.31 | 0.062; 10.2 | 0.043; 10.08 |
| BBSL-soft | NBVS | 0.02; 7.85 | 0.014; 8.43 | 0.01; 7.56 | 0.075; 8.99 | 0.058; 8.37 | 0.04; 7.49 |
| BBSL-soft | BCTS | 0.02; 7.85 | 0.014; 8.72 | 0.01; 7.21 | 0.074; 8.61 | 0.057; 7.98 | 0.038; 6.36 |
| BBSL-soft | VS | 0.02; 8.27 | 0.014; 8.66 | 0.01; 8.07 | 0.074; 8.32 | 0.057; 7.03 | 0.038; 5.8 |
| RLLS-hard | None | 0.026; 11.45 | 0.018; 11.4 | 0.013; 10.67 | 0.096; 14.22 | 0.078; 14.09 | 0.063; 14.1 |
| RLLS-soft | None | 0.021; 8.9 | 0.014; 8.08 | 0.01; 7.77 | 0.08; 11.51 | 0.064; 11.95 | 0.045; 11.71 |
| RLLS-soft | TS | 0.021; 8.26 | 0.014; 7.22 | 0.01; 7.17 | 0.078; 10.33 | 0.062; 10.54 | 0.043; 10.46 |
| RLLS-soft | NBVS | 0.02; 7.71 | 0.014; 8.25 | 0.01; 7.42 | 0.075; 9.11 | 0.058; 8.91 | 0.04; 8.33 |
| RLLS-soft | BCTS | 0.02; 7.64 | 0.014; 8.54 | 0.01; 7.01 | 0.074; 8.77 | 0.057; 8.55 | 0.038; 7.26 |
| RLLS-soft | VS | 0.02; 8.07 | 0.014; 8.44 | 0.01; 7.91 | 0.074; 8.41 | 0.057; 7.66 | 0.038; 6.72 |

Table E.5: **CIFAR10: Comparison of all calibration and domain adaptation methods, using JS Divergence (Sec. B) as the metric ("tweak-one" shift).** Value before the semicolon is the average of the metric over all trials. Value after the semicolon is the average rank of the domain adaptation + calibration method combination relative to the other method combinations in the column. Bold values in a column are not significantly different from the best-performing method in the column as measured by a paired Wilcoxon test at $p < 0.01$. EM with BCTS or VS tends to achieve the best performance. See **Sec. 4.1** for details on the experimental setup.

| Shift Estimator | Calibration Method | $\alpha = 0.1$ | | | $\alpha = 1.0$ | | | $\alpha = 10$ | | |
|---|---|---|---|---|---|---|---|---|---|---|
| | | n=2000 | n=4000 | n=8000 | n=2000 | n=4000 | n=8000 | n=2000 | n=4000 | n=8000 |
| EM | None | **0.035; 0.28** | **0.033; 0.42** | **0.033; 0.8** | **0.021; 1.28** | **0.018; 1.76** | 0.016; 2.28 | **0.016; 1.44** | **0.011; 1.8** | 0.009; 2.2 |
| BBSL-hard | None | 0.076; 3.36 | 0.064; 3.32 | 0.055; 3.13 | 0.03; 2.65 | 0.023; 2.61 | 0.016; 2.6 | 0.019; 2.86 | 0.013; 2.52 | 0.009; 2.27 |
| BBSL-soft | None | 0.068; 2.26 | 0.056; 2.23 | 0.049; 2.22 | 0.027; 1.67 | 0.019; 1.46 | 0.014; 1.2 | 0.016; 1.62 | 0.011; 1.58 | 0.008; 1.42 |
| RLLS-hard | None | 0.074; 2.6 | 0.063; 2.49 | 0.054; 2.37 | 0.03; 2.69 | 0.023; 2.59 | 0.016; 2.62 | 0.019; 2.66 | 0.013; 2.58 | 0.009; 2.47 |
| RLLS-soft | None | 0.066; 1.5 | 0.055; 1.54 | 0.048; 1.54 | 0.027; 1.71 | 0.019; 1.58 | 0.014; 1.3 | **0.016; 1.42** | 0.011; 1.52 | **0.008; 1.64** |
| EM | TS | **0.024; 0.04** | **0.022; 0.09** | **0.021; 0.14** | **0.019; 0.58** | **0.017; 0.94** | 0.014; 1.18 | 0.016; 1.18 | 0.012; 1.22 | 0.009; 1.26 |
| BBSL-soft | TS | 0.067; 1.81 | 0.056; 1.78 | 0.048; 1.71 | 0.026; 1.21 | **0.019; 0.94** | 0.013; 0.84 | **0.016; 1.0** | 0.011; 0.93 | **0.008; 0.77** |
| RLLS-soft | TS | 0.066; 1.15 | 0.055; 1.13 | 0.048; 1.15 | 0.026; 1.21 | 0.019; 1.12 | 0.014; 1.12 | **0.016; 0.82** | 0.011; 0.85 | **0.008; 0.97** |
| EM | NBVS | **0.021; 0.02** | **0.017; 0.04** | **0.017; 0.05** | **0.017; 0.3** | **0.013; 0.38** | **0.01; 0.46** | 0.015; 0.88 | 0.01; 0.82 | 0.008; 0.96 |
| BBSL-soft | NBVS | 0.067; 1.84 | 0.055; 1.8 | 0.048; 1.8 | 0.026; 1.32 | 0.018; 1.21 | 0.013; 1.22 | 0.016; 1.15 | 0.011; 1.14 | 0.008; 0.97 |
| RLLS-soft | NBVS | 0.066; 1.14 | 0.054; 1.16 | 0.047; 1.15 | 0.026; 1.38 | 0.018; 1.41 | 0.013; 1.32 | 0.016; 0.97 | 0.011; 1.04 | 0.008; 1.07 |
| EM | BCTS | **0.02; 0.02** | **0.016; 0.02** | **0.015; 0.08** | **0.016; 0.22** | **0.012; 0.34** | **0.009; 0.36** | **0.015; 0.84** | **0.01; 0.76** | 0.008; 0.86 |
| BBSL-soft | BCTS | 0.066; 1.82 | 0.055; 1.81 | 0.047; 1.75 | 0.025; 1.4 | 0.018; 1.22 | 0.014; 1.29 | 0.016; 1.18 | 0.011; 1.14 | 0.008; 1.01 |
| RLLS-soft | BCTS | 0.065; 1.16 | 0.054; 1.17 | 0.046; 1.35 | 0.025; 1.38 | 0.018; 1.44 | 0.014; 1.44 | **0.016; 0.97** | 0.011; 1.1 | 0.008; 1.13 |
| EM | VS | **0.023; 0.02** | **0.017; 0.0** | **0.015; 0.06** | **0.016; 0.3** | **0.013; 0.28** | **0.009; 0.44** | **0.015; 0.88** | **0.01; 0.72** | 0.008; 0.86 |
| BBSL-soft | VS | 0.066; 1.82 | 0.055; 1.86 | 0.047; 1.76 | 0.025; 1.34 | 0.018; 1.26 | 0.014; 1.24 | 0.016; 1.18 | 0.011; 1.17 | 0.008; 1.0 |
| RLLS-soft | VS | 0.065; 1.16 | 0.054; 1.14 | 0.046; 1.18 | 0.026; 1.36 | 0.018; 1.46 | 0.014; 1.32 | **0.016; 0.94** | 0.011; 1.11 | 0.008; 1.14 |

Table E.6: **CIFAR10: Comparison of EM, BBSL and RLLS (dirichlet shift) using JS Divergence as the metric.** Analogous to **Table 1**, but with JS Divergence as the metric rather than MSE.

| Shift Estimator | Calibration Method | $\rho = 0.01$ | | | $\rho = 0.9$ | | |
|---|---|---|---|---|---|---|---|
| | | $n$=2000 | $n$=4000 | $n$=8000 | $n$=2000 | $n$=4000 | $n$=8000 |
| EM | None | **0.018; 1.32** | **0.013; 1.66** | 0.012; 2.6 | **0.047; 0.14** | **0.045; 0.56** | **0.044; 1.42** |
| BBSL-hard | None | 0.026; 2.81 | 0.018; 2.78 | 0.013; 2.55 | 0.096; 3.2 | 0.078; 2.87 | 0.063; 2.82 |
| BBSL-soft | None | 0.021; 1.61 | 0.014; 1.62 | 0.01; 1.4 | 0.08; 1.8 | 0.064; 1.5 | **0.045; 1.03** |
| RLLS-hard | None | 0.026; 2.73 | 0.018; 2.52 | 0.013; 2.21 | 0.096; 3.18 | 0.078; 3.25 | 0.063; 3.16 |
| RLLS-soft | None | **0.021; 1.53** | **0.014; 1.42** | **0.01; 1.24** | 0.08; 1.68 | 0.064; 1.82 | 0.045; 1.57 |
| EM | TS | **0.02; 0.98** | 0.015; 1.1 | 0.014; 1.46 | **0.036; 0.04** | **0.034; 0.14** | **0.032; 0.48** |
| BBSL-soft | TS | 0.021; 1.05 | 0.014; 1.06 | 0.01; 0.88 | 0.078; 1.46 | 0.062; 1.25 | 0.043; 1.07 |
| RLLS-soft | TS | **0.021; 0.97** | **0.014; 0.84** | **0.01; 0.66** | 0.078; 1.5 | 0.062; 1.61 | 0.043; 1.45 |
| EM | NBVS | **0.016; 0.56** | **0.01; 0.6** | **0.008; 0.76** | **0.033; 0.04** | **0.028; 0.18** | **0.026; 0.42** |
| BBSL-soft | NBVS | 0.02; 1.28 | 0.014; 1.29 | 0.01; 1.19 | 0.075; 1.45 | 0.058; 1.12 | 0.04; 0.86 |
| RLLS-soft | NBVS | 0.02; 1.16 | 0.014; 1.11 | 0.01; 1.05 | 0.075; 1.51 | 0.058; 1.7 | 0.04; 1.72 |
| EM | BCTS | **0.016; 0.68** | **0.01; 0.54** | **0.009; 0.78** | **0.028; 0.0** | **0.023; 0.1** | **0.021; 0.24** |
| BBSL-soft | BCTS | 0.02; 1.24 | 0.014; 1.32 | 0.01; 1.21 | 0.074; 1.44 | 0.057; 1.17 | 0.038; 0.92 |
| RLLS-soft | BCTS | 0.02; 1.08 | 0.014; 1.14 | 0.01; 1.01 | 0.074; 1.56 | 0.057; 1.73 | 0.038; 1.84 |
| EM | VS | **0.016; 0.58** | **0.01; 0.42** | **0.008; 0.54** | **0.029; 0.02** | **0.023; 0.08** | **0.02; 0.1** |
| BBSL-soft | VS | 0.02; 1.29 | 0.014; 1.4 | 0.01; 1.31 | 0.074; 1.46 | 0.057; 1.15 | 0.038; 0.98 |
| RLLS-soft | VS | 0.02; 1.13 | 0.014; 1.18 | 0.01; 1.15 | 0.074; 1.52 | 0.057; 1.77 | 0.038; 1.92 |

Table E.7: **CIFAR10: Comparison of EM, BBSL and RLLS ("tweak-one" shift) using JS Divergence as the metric.** Analogous to **Table 1**, but with tweak-one shift instead of dirichlet shift and JS Divergence as the metric rather than MSE.

| Shift Estimator | Calibration Method | $\rho = 0.01$ | | | $\rho = 0.9$ | | |
|---|---|---|---|---|---|---|---|
| | | $n$=2000 | $n$=4000 | $n$=8000 | $n$=2000 | $n$=4000 | $n$=8000 |
| EM | None | 0.00202; 1.8 | 0.00104; 2.38 | 0.00069; 3.04 | 0.08233; 2.83 | 0.07773; 3.23 | 0.0756; 3.74 |
| BBSL-hard | None | 0.00258; 2.95 | 0.00119; 2.77 | 0.00059; 2.53 | **0.02157; 2.04** | 0.01198; 1.83 | **0.00599; 1.37** |
| BBSL-soft | None | 0.00183; 1.45 | 0.00081; 1.31 | 0.00045; 1.33 | **0.01905; 1.47** | 0.01102; 1.19 | 0.00612; 1.2 |
| RLLS-hard | None | 0.00256; 2.68 | 0.00119; 2.47 | 0.00059; 2.11 | 0.02055; 2.03 | 0.01172; 2.09 | **0.00596; 1.74** |
| RLLS-soft | None | **0.00182; 1.12** | **0.00081; 1.07** | **0.00045; 0.99** | 0.01872; 1.63 | 0.011; 1.66 | 0.0061; 1.95 |
| EM | TS | 0.00212; 1.35 | 0.00111; 1.5 | 0.00074; 1.82 | 0.11247; 1.42 | 0.10919; 1.48 | 0.10942; 1.6 |
| BBSL-soft | TS | 0.00181; 0.96 | 0.0008; 0.89 | 0.00044; 0.74 | **0.0201; 0.6** | 0.01171; 0.46 | 0.00677; 0.3 |
| RLLS-soft | TS | **0.00179; 0.69** | **0.0008; 0.61** | **0.00044; 0.44** | **0.0198; 0.98** | 0.0117; 1.06 | 0.00675; 1.1 |
| EM | NBVS | **0.0015; 0.62** | **0.00066; 0.66** | **0.00041; 0.82** | 0.0041; 0.18 | 0.00263; 0.42 | 0.00206; 0.64 |
| BBSL-soft | NBVS | 0.00175; 1.42 | 0.0008; 1.36 | 0.00047; 1.28 | 0.01791; 1.25 | 0.00921; 0.98 | 0.00498; 0.75 |
| RLLS-soft | NBVS | 0.00174; 0.96 | 0.0008; 0.98 | 0.00047; 0.9 | 0.01757; 1.57 | 0.00916; 1.6 | 0.00496; 1.61 |
| EM | BCTS | **0.0015; 0.6** | **0.00065; 0.58** | **0.0004; 0.6** | **0.00221; 0.14** | **0.00137; 0.16** | **0.00104; 0.18** |
| BBSL-soft | BCTS | 0.00174; 1.41 | 0.00081; 1.39 | 0.00046; 1.38 | 0.01808; 1.28 | 0.00937; 1.13 | 0.00501; 0.96 |
| RLLS-soft | BCTS | 0.00173; 0.99 | 0.00081; 1.03 | 0.00046; 1.02 | 0.01769; 1.58 | 0.0093; 1.71 | 0.00499; 1.86 |
| EM | VS | **0.00151; 0.56** | **0.00064; 0.52** | **0.0004; 0.56** | **0.00266; 0.04** | **0.00144; 0.12** | **0.00104; 0.22** |
| BBSL-soft | VS | 0.00176; 1.46 | 0.00081; 1.43 | 0.00047; 1.43 | 0.01798; 1.29 | 0.00912; 1.13 | 0.00486; 0.96 |
| RLLS-soft | VS | 0.00175; 0.98 | 0.00081; 1.05 | 0.00047; 1.01 | 0.01759; 1.67 | 0.00906; 1.75 | 0.00484; 1.82 |

Table E.8: **CIFAR10: Comparison of EM, BBSL and RLLS ("tweak-one" shift) using MSE as the metric.** Analogous to **Table 1**, but with tweak-one shift instead of dirichlet shift.

# F MNIST TABLES

| Shift Estimator | Calibration Method | $\alpha = 0.1$ | | | $\alpha = 1.0$ | | | $\alpha = 10$ | | |
|---|---|---|---|---|---|---|---|---|---|---|
| | | n=2000 | n=4000 | n=8000 | n=2000 | n=4000 | n=8000 | n=2000 | n=4000 | n=8000 |
| EM | None | 0.01046; 9.61 | 0.00786; 8.99 | 0.00587; 9.41 | 0.00484; 12.34 | 0.0034; 13.06 | 0.00328; 13.82 | 0.00262; 12.45 | 0.00143; 12.85 | 0.00101; 13.85 |
| EM | TS | 0.00945; 8.17 | 0.00658; 7.88 | 0.00476; 8.82 | 0.00265; 7.55 | 0.00142; 7.13 | 0.00117; 8.46 | 0.00193; 7.55 | 0.00089; 7.1 | 0.00057; 7.06 |
| EM | NBVS | 0.00243; 3.81 | 0.00187; 3.94 | 0.00143; 4.3 | **0.00202; 4.6** | **0.00107; 4.17** | **0.0079; 5.4** | 0.00193; 7.77 | 0.00088; 7.14 | 0.00056; 6.74 |
| EM | BCTS | **0.00128; 1.82** | **0.00094; 1.97** | 0.00088; 2.73 | **0.00191; 4.07** | **0.00107; 4.43** | 0.0008; 5.85 | 0.00187; 6.29 | **0.00084; 5.92** | **0.00054; 5.73** |
| EM | VS | **0.00133; 1.85** | **0.00093; 1.98** | **0.00083; 2.23** | 0.00195; 4.46 | **0.00107; 4.64** | 0.0008; 5.92 | 0.00194; 7.74 | **0.00086; 6.87** | 0.00056; 7.22 |
| BBSL-hard | None | 0.00688; 10.4 | 0.00467; 10.04 | 0.0033; 10.3 | 0.00326; 10.98 | 0.00201; 11.63 | 0.00134; 11.34 | 0.00228; 11.28 | 0.00114; 11.15 | 0.00082; 12.51 |
| BBSL-soft | None | 0.00634; 9.95 | 0.0043; 10.5 | 0.0029; 9.92 | 0.00274; 8.96 | 0.00151; 8.91 | 0.00101; 8.27 | 0.00188; 8.51 | 0.00092; 8.89 | 0.00056; 7.76 |
| BBSL-soft | TS | 0.00573; 9.07 | 0.00385; 9.26 | 0.00262; 8.35 | 0.00261; 7.2 | 0.00144; 6.61 | 0.00096; 6.79 | 0.0018; 5.96 | **0.00086; 6.28** | **0.00054; 5.58** |
| BBSL-soft | NBVS | 0.00629; 9.78 | 0.00397; 9.69 | 0.00269; 9.42 | 0.00265; 8.51 | 0.00146; 7.87 | 0.00096; 7.15 | 0.00187; 7.86 | 0.00088; 7.92 | 0.00055; 7.44 |
| BBSL-soft | BCTS | 0.00612; 9.53 | 0.00392; 9.41 | 0.00269; 9.2 | 0.00262; 7.51 | 0.00145; 7.26 | 0.00095; 6.32 | 0.00183; 6.4 | **0.00086; 6.64** | **0.00054; 6.2** |
| BBSL-soft | VS | 0.00635; 9.36 | 0.0041; 9.66 | 0.00394; 9.3 | 0.00265; 8.62 | 0.00149; 8.72 | 0.00098; 8.39 | 0.00187; 7.8 | 0.00088; 7.48 | 0.00056; 8.64 |
| RLLS-hard | None | 0.00666; 10.03 | 0.00455; 9.69 | 0.00324; 9.74 | 0.00325; 10.79 | 0.00201; 11.5 | 0.00134; 11.24 | 0.00228; 10.86 | 0.00114; 11.03 | 0.00082; 12.29 |
| RLLS-soft | None | 0.00613; 8.94 | 0.00373; 9.66 | 0.00283; 9.13 | 0.00274; 8.84 | 0.00151; 9.07 | 0.00101; 8.28 | 0.00188; 8.39 | 0.00092; 8.75 | 0.00056; 7.54 |
| RLLS-soft | TS | 0.00558; 8.12 | 0.00373; 8.28 | 0.00258; 7.51 | 0.00261; 7.1 | 0.00144; 6.67 | 0.00096; 6.79 | **0.0018; 5.7** | **0.00086; 6.14** | **0.00054; 5.46** |
| RLLS-soft | NBVS | 0.00617; 8.76 | 0.00385; 8.66 | 0.00265; 8.51 | 0.00265; 8.42 | 0.00146; 8.0 | 0.00096; 7.18 | 0.00187; 7.62 | 0.00088; 7.84 | 0.00055; 7.36 |
| RLLS-soft | BCTS | 0.00599; 8.39 | 0.0038; 8.32 | 0.00265; 8.27 | 0.00262; 7.44 | 0.00145; 7.45 | 0.00095; 6.37 | 0.00183; 6.2 | **0.00086; 6.58** | **0.00054; 6.06** |
| RLLS-soft | VS | 0.00623; 8.41 | 0.00384; 8.43 | 0.00266; 8.65 | 0.00265; 8.61 | 0.00149; 8.88 | 0.00098; 8.43 | 0.00189; 7.62 | 0.00088; 7.42 | 0.00056; 8.56 |

Table F.1: **MNIST: Comparison of all calibration and domain adaptation methods, using MSE (Sec. 3.3) as the metric (dirichlet shift).** Value before the semicolon is the average of the metric over all trials. Value after the semicolon is the average rank of the domain adaptation + calibration method combination relative to the other method combinations in the column. Bold values in a column are not significantly different from the best-performing method in the column as measured by a paired Wilcoxon test at $p < 0.01$. EM with BCTS or VS tends to achieve the best performance, particularly for larger amounts of shift (corresponding to smaller $\alpha$). See **Sec. 4.1** for details on the experimental setup.

| Shift Estimator | Calibration Method | $\rho = 0.01$ | | | $\rho = 0.9$ | | |
|---|---|---|---|---|---|---|---|
| | | n=2000 | n=4000 | n=8000 | n=2000 | n=4000 | n=8000 |
| EM | None | 0.00219; 11.93 | 0.00112; 11.76 | 0.00072; 12.0 | 0.00998; 9.46 | 0.00648; 8.47 | 0.00528; 9.71 |
| EM | TS | 0.00183; 7.28 | 0.00091; 6.61 | 0.00058; 5.55 | 0.0062; 6.44 | 0.00325; 5.72 | 0.00199; 4.86 |
| EM | NBVS | 0.00177; 6.01 | 0.00088; 5.53 | 0.00056; 4.3 | 0.00181; 2.45 | 0.00088; 2.47 | 0.00044; 1.8 |
| EM | BCTS | **0.00173; 5.1** | **0.00087; 5.07** | **0.00056; 3.74** | **0.0007; 0.75** | **0.00043; 0.69** | **0.0003; 0.61** |
| EM | VS | 0.00177; 6.18 | 0.00087; 5.01 | **0.00056; 3.71** | 0.00099; 1.0 | 0.00049; 1.29 | 0.00033; 1.33 |
| BBSL-hard | None | 0.00235; 11.89 | 0.00123; 12.7 | 0.00083; 14.18 | 0.01183; 12.81 | 0.00796; 13.47 | 0.00652; 14.83 |
| BBSL-soft | None | 0.00186; 9.08 | 0.00099; 9.18 | 0.00063; 9.44 | 0.00926; 10.52 | 0.00488; 8.26 | 0.00336; 5.59 |
| BBSL-soft | TS | 0.00178; 6.63 | 0.00093; 6.6 | 0.0006; 6.73 | 0.00914; 10.29 | 0.00515; 9.41 | 0.00384; 9.25 |
| BBSL-soft | NBVS | 0.00184; 8.63 | 0.00096; 8.7 | 0.00062; 8.93 | 0.00887; 9.19 | 0.00509; 8.99 | 0.0038; 8.79 |
| BBSL-soft | BCTS | 0.0018; 6.94 | 0.00094; 7.3 | 0.00061; 6.66 | 0.00879; 9.18 | 0.00506; 8.93 | 0.00373; 7.69 |
| BBSL-soft | VS | 0.00184; 8.57 | 0.00096; 8.55 | 0.00063; 10.07 | 0.00894; 9.96 | 0.00526; 10.98 | 0.00415; 12.5 |
| RLLS-hard | None | 0.00235; 11.15 | 0.00123; 11.86 | 0.00083; 13.2 | 0.01099; 11.75 | 0.0076; 12.84 | 0.00633; 14.05 |
| RLLS-soft | None | 0.00186; 8.42 | 0.00099; 8.5 | 0.00063; 8.58 | 0.00875; 9.3 | 0.00478; 7.95 | 0.00335; 5.92 |
| RLLS-soft | TS | 0.00178; 5.89 | 0.00093; 5.96 | 0.0006; 5.85 | 0.00863; 9.0 | 0.00505; 8.94 | 0.00383; 9.41 |
| RLLS-soft | NBVS | 0.00184; 8.01 | 0.00096; 8.08 | 0.00062; 8.05 | 0.0084; 7.74 | 0.00499; 8.59 | 0.00379; 8.94 |
| RLLS-soft | BCTS | 0.0018; 6.28 | 0.00094; 6.66 | 0.00061; 5.78 | 0.00832; 7.83 | 0.00497; 8.54 | 0.00372; 8.0 |
| RLLS-soft | VS | 0.00184; 8.01 | 0.00096; 7.93 | 0.00063; 9.23 | 0.00843; 8.33 | 0.00515; 10.46 | 0.00413; 12.72 |

Table F.2: **MNIST: Comparison of all calibration and domain adaptation methods, using MSE (Sec. 3.3) as the metric ("tweak-one" shift).** Value before the semicolon is the average of the metric over all trials. Value after the semicolon is the average rank of the domain adaptation + calibration method combination relative to the other method combinations in the column. Bold values in a column are not significantly different from the best-performing method in the column as measured by a paired Wilcoxon test at $p < 0.01$. EM with BCTS or VS tends to achieve the best performance. See **Sec. 4.1** for details on the experimental setup.

| Shift Estimator | Calibration Method | $\alpha = 0.1$ | | | $\alpha = 1.0$ | | | $\alpha = 10$ | | |
|---|---|---|---|---|---|---|---|---|---|---|
| | | n=2000 | n=4000 | n=8000 | n=2000 | n=4000 | n=8000 | n=2000 | n=4000 | n=8000 |
| EM | None | 0.01046; 2.06 | 0.00786; 1.86 | 0.00587; 2.02 | 0.00484; 2.88 | 0.0034; 3.04 | 0.00328; 3.3 | 0.00262; 2.74 | 0.00143; 2.74 | 0.00101; 3.0 |
| BBSL-hard | None | 0.00688; 2.34 | 0.00467; 2.13 | 0.0033; 2.32 | 0.00326; 2.15 | 0.00201; 2.31 | 0.00134; 2.14 | 0.00228; 2.57 | 0.00114; 2.32 | 0.00082; 2.55 |
| BBSL-soft | None | 0.00634; 1.97 | 0.0043; 2.25 | 0.0029; 2.05 | **0.00274; 1.54** | **0.00151; 1.15** | **0.00101; 1.25** | 0.00188; 1.33 | **0.00092; 1.44** | **0.00056; 1.17** |
| RLLS-hard | None | 0.00666; 2.04 | 0.00455; 1.93 | 0.00324; 1.92 | 0.00325; 1.95 | 0.00201; 2.19 | 0.00134; 2.04 | 0.00228; 2.15 | 0.00114; 2.2 | 0.00082; 2.33 |
| RLLS-soft | None | **0.00613; 1.59** | **0.0041; 1.83** | **0.00283; 1.69** | **0.00274; 1.48** | **0.00151; 1.31** | **0.00101; 1.27** | **0.00188; 1.21** | **0.00092; 1.3** | **0.00056; 0.95** |
| EM | TS | 0.00945; 0.91 | 0.00658; 0.87 | 0.00476; 1.0 | 0.00265; 0.92 | 0.00142; 0.9 | 0.00117; 1.14 | 0.00193; 1.16 | 0.00089; 1.02 | 0.00057; 1.08 |
| BBSL-soft | TS | 0.00573; 1.18 | 0.00385; 1.21 | 0.00262; 1.16 | 0.00261; 1.07 | 0.00144; 1.02 | 0.00096; 0.93 | 0.0018; 1.05 | 0.00086; 1.06 | 0.00054; 1.02 |
| RLLS-soft | TS | **0.00558; 0.91** | **0.00373; 0.92** | **0.00258; 0.84** | 0.00261; 1.01 | 0.00144; 1.08 | 0.00096; 0.93 | **0.0018; 0.79** | 0.00086; 0.92 | 0.00054; 0.9 |
| EM | NBVS | **0.00243; 0.36** | **0.00187; 0.38** | **0.00143; 0.44** | **0.00202; 0.52** | **0.00107; 0.5** | **0.0079; 0.86** | 0.00193; 1.0 | **0.00088; 0.92** | **0.00056; 0.9** |
| BBSL-soft | NBVS | 0.00629; 1.46 | 0.00397; 1.47 | 0.00269; 1.44 | 0.00265; 1.28 | 0.00146; 1.18 | 0.00096; 1.05 | 0.00187; 1.12 | 0.00088; 1.08 | 0.00055; 1.09 |
| RLLS-soft | NBVS | 0.00617; 1.18 | 0.00385; 1.15 | 0.00265; 1.12 | 0.00265; 1.2 | 0.00146; 1.32 | 0.00096; 1.09 | 0.00187; 0.88 | 0.00088; 1.0 | 0.00055; 1.01 |
| EM | BCTS | **0.00128; 0.12** | **0.00094; 0.18** | **0.00088; 0.28** | **0.00191; 0.48** | **0.00107; 0.54** | 0.0008; 0.82 | 0.00187; 0.98 | **0.00084; 0.94** | **0.00054; 0.9** |
| BBSL-soft | BCTS | 0.00612; 1.6 | 0.00392; 1.55 | 0.00269; 1.51 | 0.00262; 1.29 | 0.00145; 1.15 | 0.00095; 1.06 | 0.00183; 1.11 | 0.00086; 1.12 | 0.00054; 1.12 |
| RLLS-soft | BCTS | 0.00599; 1.28 | 0.0038; 1.27 | 0.00265; 1.21 | 0.00262; 1.23 | 0.00145; 1.33 | 0.00095; 1.12 | **0.00183; 0.91** | 0.00086; 1.0 | 0.00054; 0.98 |
| EM | VS | **0.00133; 0.14** | **0.00093; 0.17** | **0.00083; 0.19** | **0.00195; 0.52** | **0.00107; 0.51** | 0.0008; 0.78 | 0.00194; 0.96 | **0.00086; 0.84** | **0.00056; 0.88** |
| BBSL-soft | VS | 0.00635; 1.55 | 0.00394; 1.56 | 0.0027; 1.56 | 0.00265; 1.22 | 0.00149; 1.17 | 0.00098; 1.09 | 0.00189; 1.11 | 0.00088; 1.11 | 0.00056; 1.1 |
| RLLS-soft | VS | 0.00623; 1.31 | 0.00384; 1.27 | 0.00266; 1.25 | 0.00265; 1.26 | 0.00149; 1.32 | 0.00098; 1.13 | **0.00189; 0.93** | 0.00088; 1.05 | 0.00056; 1.02 |

Table F.3: **MNIST: Comparison of EM, BBSL and RLLS (dirichlet shift).** Analogous to **Table 2**, but with dirichlet shift rather than tweak-one shift.

# G    CIFAR100 SUPPLEMENTARY TABLES

| Shift Estimator | Calibration Method | $\alpha = 0.1$ | | | $\alpha = 1.0$ | | | $\alpha = 10.0$ | | |
|---|---|---|---|---|---|---|---|---|---|---|
| | | $n=7000$ | $n=8500$ | $n=10000$ | $n=7000$ | $n=8500$ | $n=10000$ | $n=7000$ | $n=8500$ | $n=10000$ |
| EM | None | 2.26413; 13.56 | 2.13137; 13.96 | 2.08096; 14.13 | 0.75139; 15.04 | 0.6941; 15.36 | 0.66819; 15.39 | 0.41269; 15.64 | 0.38438; 15.85 | 0.36558; 15.94 |
| EM | TS | 0.85732; 8.05 | 0.73074; 8.15 | 0.65051; 7.88 | 0.34451; 9.95 | 0.30896; 10.44 | 0.28795; 10.42 | 0.17923; 10.09 | 0.15609; 10.56 | 0.14494; 11.42 |
| EM | NBVS | 0.28904; 4.03 | 0.27676; 4.07 | 0.26944; 4.48 | 0.15848; 4.75 | 0.14828; 5.34 | 0.14304; 6.21 | 0.11329; 5.4 | 0.10635; 7.03 | 0.10256; 8.17 |
| EM | BCTS | 0.2458; 3.16 | 0.25185; 3.53 | 0.25628; 3.48 | 0.14527; 3.61 | 0.14006; 4.37 | 0.13766; 5.31 | 0.10338; 3.74 | 0.09803; 5.44 | 0.09538; 6.54 |
| EM | VS | **0.1994; 2.41** | **0.2011; 2.67** | **0.20436; 2.67** | **0.13788; 3.06** | **0.1307; 3.62** | **0.12736; 4.34** | **0.10468; 4.02** | 0.09869; 5.62 | 0.09667; 6.88 |
| BBSL-hard | None | 1.7799; 13.66 | 1.27283; 13.5 | 1.19495; 13.68 | 0.4737; 14.31 | 0.39212; 14.13 | 0.35386; 14.32 | 0.2997; 14.84 | 0.24168; 14.75 | 0.2161; 14.65 |
| BBSL-soft | None | 1.32248; 11.19 | 0.94221; 11.34 | 0.83588; 11.8 | 0.32731; 11.94 | 0.27302; 12.09 | 0.24683; 12.19 | 0.21342; 12.63 | 0.16996; 12.46 | 0.14857; 12.35 |
| BBSL-soft | TS | 1.0511; 8.67 | 0.73046; 8.85 | 0.61651; 8.4 | 0.25082; 8.81 | 0.20128; 8.56 | 0.17385; 8.13 | 0.15657; 7.99 | 0.11901; 7.15 | 0.10114; 6.49 |
| BBSL-soft | NBVS | 1.01696; 8.98 | 0.69643; 8.76 | 0.60503; 8.65 | 0.24203; 8.3 | 0.19391; 8.27 | 0.16837; 8.11 | 0.15685; 8.48 | 0.12001; 7.56 | 0.10221; 7.43 |
| BBSL-soft | BCTS | 0.97278; 8.68 | 0.68114; 8.64 | 0.59169; 8.52 | 0.24328; 8.36 | 0.19399; 8.06 | 0.16944; 8.06 | 0.15524; 8.14 | 0.11855; 7.11 | 0.10079; 6.77 |
| BBSL-soft | VS | 0.94791; 8.61 | 0.66421; 8.38 | 0.57766; 8.3 | 0.23665; 7.55 | 0.18917; 7.44 | 0.16374; 7.11 | 0.1519; 7.32 | 0.116; 6.29 | 0.09866; 5.76 |
| RLLS-hard | None | 0.89184; 10.2 | 0.74954; 10.58 | 0.71544; 10.44 | 0.31391; 10.82 | 0.26279; 10.74 | 0.23164; 10.8 | 0.18167; 10.78 | 0.15771; 11.1 | 0.14006; 10.62 |
| RLLS-soft | None | 0.73652; 8.03 | 0.61146; 7.78 | 0.57115; 8.32 | 0.22919; 7.92 | 0.19488; 7.86 | 0.17308; 8.2 | 0.1429; 8.06 | 0.12089; 8.13 | 0.1065; 7.67 |
| RLLS-soft | TS | 0.70936; 6.68 | 0.58352; 6.61 | 0.52749; 6.46 | 0.20268; 5.93 | 0.1665; 5.45 | 0.14336; 4.91 | 0.12306; 5.39 | 0.09967; 4.93 | 0.08668; 4.41 |
| RLLS-soft | NBVS | 0.65047; 6.8 | 0.52242; 6.39 | 0.48347; 6.25 | 0.19225; 5.26 | 0.15747; 4.91 | **0.13543; 4.37** | 0.12045; 4.73 | 0.09735; 4.44 | 0.08459; 4.09 |
| RLLS-soft | BCTS | 0.63399; 6.45 | 0.51168; 6.17 | 0.47275; 6.04 | 0.19027; 5.05 | 0.15529; 4.33 | **0.1341; 3.99** | 0.11849; 4.1 | **0.09528; 3.38** | **0.08269; 3.01** |
| RLLS-soft | VS | 0.64403; 6.84 | 0.52134; 6.62 | 0.47947; 6.5 | 0.1941; 5.34 | 0.15799; 5.03 | **0.1352; 4.14** | 0.11968; 4.65 | 0.09656; 4.2 | 0.08386; 3.8 |

Table G.1: **CIFAR100: Comparison of all calibration and domain adaptation methods, using MSE (Sec. 3.3) as the metric (dirichlet shift).** Value before the semicolon is the average of the metric over all trials. Value after the semicolon is the average rank of the domain adaptation + calibration method combination relative to the other method combinations in the column. Bold values in a column are not significantly different from the best-performing method in the column as measured by a paired Wilcoxon test at $p < 0.01$. EM with VS tends to achieve the best performance, particularly for larger amounts of shift (corresponding to smaller $\alpha$). See **Sec. 4.1** for details on the experimental setup.

| Shift Estimator | Calibration Method | $\alpha = 0.1$ | | | $\alpha = 1.0$ | | | $\alpha = 10.0$ | | |
|---|---|---|---|---|---|---|---|---|---|---|
| | | $n=7000$ | $n=8500$ | $n=10000$ | $n=7000$ | $n=8500$ | $n=10000$ | $n=7000$ | $n=8500$ | $n=10000$ |
| EM | None | 0.233; 11.7 | 0.232; 12.11 | 0.232; 12.58 | 0.232; 15.17 | 0.231; 15.44 | 0.23; 15.58 | 0.235; 15.81 | 0.234; 15.89 | 0.233; 15.94 |
| EM | TS | **0.113; 1.27** | **0.113; 1.37** | **0.113; 1.45** | **0.109; 1.07** | **0.107; 1.05** | **0.106; 1.14** | **0.107; 0.96** | **0.105; 0.87** | **0.103; 0.92** |
| EM | NBVS | 0.119; 2.37 | 0.119; 2.43 | 0.119; 2.48 | 0.118; 2.68 | 0.117; 2.86 | 0.116; 3.13 | 0.117; 2.92 | 0.115; 3.04 | 0.113; 3.21 |
| EM | BCTS | 0.118; 1.99 | 0.117; 2.12 | 0.117; 2.12 | 0.112; 1.53 | 0.111; 1.78 | 0.11; 1.76 | 0.112; 1.63 | 0.11; 1.6 | 0.108; 1.8 |
| EM | VS | **0.113; 1.53** | **0.112; 1.46** | **0.112; 1.57** | **0.111; 1.21** | **0.108; 1.06** | **0.107; 1.27** | **0.108; 0.9** | **0.106; 0.86** | **0.104; 0.91** |
| BBSL-hard | None | 0.278; 15.89 | 0.27; 15.83 | 0.267; 15.88 | 0.239; 15.57 | 0.232; 15.47 | 0.227; 15.36 | 0.221; 15.19 | 0.215; 15.1 | 0.209; 15.06 |
| BBSL-soft | None | 0.248; 14.35 | 0.241; 14.31 | 0.237; 14.34 | 0.21; 14.01 | 0.204; 13.95 | 0.198; 13.91 | 0.196; 13.84 | 0.19; 13.8 | 0.183; 13.82 |
| BBSL-soft | TS | 0.224; 10.6 | 0.218; 10.83 | 0.214; 10.88 | 0.187; 10.56 | 0.181; 10.72 | 0.176; 10.83 | 0.176; 10.43 | 0.17; 10.54 | 0.165; 10.67 |
| BBSL-soft | NBVS | 0.226; 12.0 | 0.22; 11.95 | 0.216; 11.76 | 0.189; 11.49 | 0.183; 11.46 | 0.178; 11.59 | 0.178; 11.73 | 0.172; 11.86 | 0.166; 11.84 |
| BBSL-soft | BCTS | 0.224; 10.99 | 0.218; 10.97 | 0.214; 10.75 | 0.187; 10.31 | 0.181; 10.06 | 0.176; 10.11 | 0.176; 10.05 | 0.17; 10.07 | 0.164; 10.03 |
| BBSL-soft | VS | 0.226; 11.7 | 0.22; 11.59 | 0.215; 11.35 | 0.188; 10.84 | 0.182; 10.82 | 0.177; 10.76 | 0.177; 11.21 | 0.171; 11.0 | 0.165; 11.07 |
| RLLS-hard | None | 0.226; 11.48 | 0.22; 11.13 | 0.218; 11.33 | 0.195; 11.91 | 0.189; 11.83 | 0.183; 11.64 | 0.182; 11.67 | 0.176; 11.65 | 0.17; 11.55 |
| RLLS-soft | None | 0.202; 8.09 | 0.197; 8.03 | 0.193; 8.21 | 0.171; 8.04 | 0.166; 8.13 | 0.16; 8.09 | 0.161; 7.92 | 0.156; 7.99 | 0.149; 8.0 |
| RLLS-soft | TS | 0.182; 4.96 | 0.176; 4.92 | 0.172; 4.72 | 0.152; 4.9 | 0.146; 4.82 | 0.14; 4.55 | 0.146; 4.94 | 0.14; 4.81 | 0.134; 4.59 |
| RLLS-soft | NBVS | 0.186; 6.01 | 0.179; 6.05 | 0.176; 5.82 | 0.155; 6.05 | 0.149; 6.03 | 0.144; 5.92 | 0.149; 6.04 | 0.142; 6.17 | 0.137; 6.15 |
| RLLS-soft | BCTS | 0.183; 5.0 | 0.177; 5.01 | 0.174; 4.98 | 0.153; 4.74 | 0.146; 4.66 | 0.141; 4.44 | 0.146; 4.52 | 0.14; 4.46 | 0.134; 4.32 |
| RLLS-soft | VS | 0.186; 6.07 | 0.179; 5.89 | 0.176; 5.78 | 0.155; 5.92 | 0.149; 5.86 | 0.144; 5.92 | 0.149; 6.27 | 0.143; 6.29 | 0.137; 6.12 |

Table G.2: **CIFAR100: Comparison of all calibration and domain adaptation methods, using JS Divergence (Sec. B) as the metric (dirichlet shift).** Value before the semicolon is the average of the metric over all trials. Value after the semicolon is the average rank of the domain adaptation + calibration method combination relative to the other method combinations in the column. Bold values in a column are not significantly different from the best-performing method in the column as measured by a paired Wilcoxon test at $p < 0.01$. Although TS attains good performance according to the JS Divergence metric, this is not the case with the MSE metric (**Table G.1**). See **Sec. 4.1** for details on the experimental setup.

| Shift Estimator | Calibration Method | $\alpha = 0.1$ | | | $\alpha = 1.0$ | | | $\alpha = 10.0$ | | |
|---|---|---|---|---|---|---|---|---|---|---|
| | | $n=7000$ | $n=8500$ | $n=10000$ | $n=7000$ | $n=8500$ | $n=10000$ | $n=7000$ | $n=8500$ | $n=10000$ |
| EM | None | 0.233; 1.77 | 0.232; 1.96 | 0.232; 2.04 | 0.232; 3.23 | 0.231; 3.44 | 0.23; 3.58 | 0.235; 3.81 | 0.234; 3.89 | 0.233; 3.94 |
| BBSL-hard | None | 0.278; 3.93 | 0.27; 3.89 | 0.267; 3.9 | 0.239; 3.57 | 0.232; 3.47 | 0.227; 3.36 | 0.221; 3.19 | 0.215; 3.1 | 0.209; 3.06 |
| BBSL-soft | None | 0.248; 2.5 | 0.241; 2.47 | 0.237; 2.4 | 0.21; 2.01 | 0.204; 1.95 | 0.198; 1.91 | 0.196; 1.81 | 0.19; 1.8 | 0.183; 1.82 |
| RLLS-hard | None | 0.226; 1.52 | 0.22; 1.4 | 0.218; 1.44 | 0.195; 1.19 | 0.189; 1.12 | 0.183; 1.13 | 0.182; 1.18 | 0.176; 1.19 | 0.17; 1.17 |
| RLLS-soft | None | **0.202; 0.28** | **0.197; 0.28** | **0.193; 0.22** | **0.171; 0.0** | **0.166; 0.02** | **0.16; 0.02** | **0.161; 0.01** | **0.156; 0.02** | **0.149; 0.01** |
| EM | TS | **0.113; 0.03** | **0.113; 0.05** | **0.113; 0.07** | **0.109; 0.01** | **0.107; 0.02** | **0.106; 0.05** | **0.107; 0.02** | **0.105; 0.0** | **0.103; 0.02** |
| BBSL-soft | TS | 0.224; 2.0 | 0.218; 2.0 | 0.214; 2.0 | 0.187; 2.0 | 0.181; 2.0 | 0.176; 2.0 | 0.176; 2.0 | 0.17; 2.0 | 0.165; 2.0 |
| RLLS-soft | TS | 0.182; 0.97 | 0.176; 0.95 | 0.172; 0.93 | 0.152; 0.99 | 0.146; 0.98 | 0.14; 0.95 | 0.146; 0.98 | 0.14; 1.0 | 0.134; 0.98 |
| EM | NBVS | **0.119; 0.09** | **0.119; 0.09** | **0.119; 0.11** | **0.118; 0.04** | **0.117; 0.07** | **0.116; 0.1** | **0.117; 0.04** | **0.115; 0.04** | **0.113; 0.09** |
| BBSL-soft | NBVS | 0.226; 1.99 | 0.22; 1.99 | 0.216; 1.99 | 0.189; 2.0 | 0.183; 2.0 | 0.178; 2.0 | 0.178; 2.0 | 0.172; 2.0 | 0.166; 2.0 |
| RLLS-soft | NBVS | 0.186; 0.92 | 0.179; 0.92 | 0.176; 0.9 | 0.155; 0.96 | 0.149; 0.93 | 0.144; 0.9 | 0.149; 0.96 | 0.142; 0.96 | 0.137; 0.91 |
| EM | BCTS | **0.118; 0.07** | **0.117; 0.09** | **0.117; 0.09** | **0.112; 0.03** | **0.111; 0.05** | **0.11; 0.07** | **0.112; 0.03** | **0.11; 0.03** | **0.108; 0.07** |
| BBSL-soft | BCTS | 0.224; 1.99 | 0.218; 1.99 | 0.214; 1.99 | 0.187; 2.0 | 0.181; 2.0 | 0.176; 2.0 | 0.176; 2.0 | 0.17; 2.0 | 0.164; 2.0 |
| RLLS-soft | BCTS | 0.183; 0.94 | 0.177; 0.92 | 0.174; 0.92 | 0.153; 0.97 | 0.146; 0.95 | 0.141; 0.91 | 0.146; 0.97 | 0.14; 0.97 | 0.134; 0.93 |
| EM | VS | **0.113; 0.06** | **0.112; 0.07** | **0.112; 0.1** | **0.111; 0.02** | **0.108; 0.03** | **0.107; 0.06** | **0.108; 0.01** | **0.106; 0.01** | **0.104; 0.03** |
| BBSL-soft | VS | 0.226; 1.99 | 0.22; 2.0 | 0.215; 1.99 | 0.188; 2.0 | 0.182; 2.0 | 0.177; 2.0 | 0.177; 2.0 | 0.171; 2.0 | 0.165; 2.0 |
| RLLS-soft | VS | 0.186; 0.95 | 0.179; 0.93 | 0.176; 0.91 | 0.155; 0.98 | 0.149; 0.97 | 0.144; 0.94 | 0.149; 0.99 | 0.143; 0.99 | 0.137; 0.97 |

Table G.3: **CIFAR100: Comparison of EM, BBSL and RLLS (dirichlet shift).** Analogous to **Table 3**, but with JS Divergence (**Sec. B**) as the metric rather than Mean Squared Error.

# H  DIABETIC RETINOPATHY SUPPLEMENTARY TABLES

| Shift Estimator | Calibration Method | $\rho = 0.5$ | | | $\rho = 0.9$ | | |
|---|---|---|---|---|---|---|---|
| | | $n$=500 | $n$=1000 | $n$=1500 | $n$=500 | $n$=1000 | $n$=1500 |
| EM | None | **1.25845; 5.32** | 0.52957; 5.93 | 0.38875; 5.26 | 0.11245; 9.65 | 0.07876; 11.25 | 0.0807; 12.07 |
| EM | TS | **1.14032; 4.96** | **0.46493; 4.89** | **0.33443; 4.34** | 0.11045; 9.48 | 0.07959; 10.42 | 0.07946; 11.7 |
| EM | NBVS | **1.18016; 6.23** | 0.54915; 5.98 | 0.39599; 6.26 | 0.16834; 10.2 | 0.1248; 11.07 | 0.12453; 12.53 |
| EM | BCTS | **1.08208; 4.54** | **0.42625; 4.02** | **0.30394; 4.55** | 0.06894; 4.33 | **0.038; 3.91** | 0.03581; 4.15 |
| EM | VS | 1.47999; 5.58 | **0.50291; 4.78** | **0.34718; 4.96** | 0.06632; 4.22 | **0.03183; 3.4** | **0.02897; 3.71** |
| BBSL-hard | None | 695.53053; 11.48 | 1087.16344; 13.27 | 1.74585; 12.34 | 370.24487; 13.22 | 284.46239; 12.8 | 0.74325; 11.67 |
| BBSL-soft | None | 12.22058; 9.3 | 1.40652; 9.88 | 0.81453; 8.88 | 1.17077; 10.73 | 0.09828; 9.93 | 0.08817; 9.31 |
| BBSL-soft | TS | 10.71961; 9.3 | 1.28572; 8.81 | 0.78213; 8.63 | 0.53585; 9.87 | 0.08911; 9.45 | 0.0714; 8.81 |
| BBSL-soft | NBVS | 18.23611; 11.1 | 2.24104; 10.13 | 1.02109; 10.32 | 2.67819; 9.7 | 0.10892; 9.01 | 0.06728; 8.07 |
| BBSL-soft | BCTS | 61.30409; 10.7 | 1.43944; 9.08 | 0.88667; 9.61 | 0.74657; 8.1 | 0.04882; 6.26 | 0.04345; 6.08 |
| BBSL-soft | VS | 14.87357; 9.91 | 1.35907; 8.96 | 0.8657; 9.08 | 0.3301; 7.25 | 0.04923; 6.34 | 0.04185; 6.0 |
| RLLS-hard | None | 2.20358; 9.09 | 1.39785; 11.65 | 1.06393; 11.24 | 0.10203; 8.01 | 0.04938; 7.69 | 0.05432; 8.04 |
| RLLS-soft | None | 1.95288; 8.04 | 0.92663; 8.85 | 0.66962; 7.83 | 0.06693; 6.08 | 0.04131; 6.27 | 0.03922; 6.4 |
| RLLS-soft | TS | 1.86605; 7.66 | 0.90546; 7.71 | 0.64625; 7.48 | 0.0692; 6.07 | 0.04575; 6.87 | 0.04036; 7.09 |
| RLLS-soft | NBVS | 1.85194; 7.38 | 0.87914; 7.36 | 0.75071; 8.55 | 0.07211; 6.81 | 0.05395; 8.1 | 0.04557; 7.5 |
| RLLS-soft | BCTS | 2.41169; 7.91 | 0.86736; 7.31 | 0.7362; 8.6 | 0.06579; 6.18 | 0.04252; 6.49 | 0.03614; 6.47 |
| RLLS-soft | VS | 2.2435; 7.5 | 0.88968; 7.39 | 0.70032; 8.07 | **0.06544; 6.1** | 0.04173; 6.74 | 0.03524; 6.4 |

Table H.1: **Kaggle Diabetic Retinopathy Detection: Comparison of all calibration and domain adaptation methods, using MSE (Sec. 3.3) as the metric.** $\rho$ represents the porportion of healthy examples in the sfhited domain; the source distribution has $\rho = 0.73$. Value before the semicolon is the average of the metric over all trials. Value after the semicolon is the average rank of the domain adaptation + calibration method combination relative to the other method combinations in the column. Bold values in a column are not significantly different from the best-performing method in the column as measured by a paired Wilcoxon test at $p < 0.01$. See **Sec. 4.1** for details on the experimental setup.

| Shift Estimator | Calibration Method | $\rho = 0.5$ | | | $\rho = 0.9$ | | |
|---|---|---|---|---|---|---|---|
| | | $n$=500 | $n$=1000 | $n$=1500 | $n$=500 | $n$=1000 | $n$=1500 |
| EM | None | **0.077; 1.96** | **0.059; 2.03** | **0.054; 2.26** | 0.111; 10.66 | 0.1; 10.84 | 0.102; 11.62 |
| EM | TS | 0.09; 3.27 | 0.068; 2.57 | 0.061; 2.74 | 0.104; 8.84 | 0.094; 8.91 | 0.094; 9.55 |
| EM | NBVS | 0.107; 5.56 | 0.089; 5.25 | 0.079; 5.87 | 0.11; 9.69 | 0.1; 9.95 | 0.102; 10.77 |
| EM | BCTS | 0.111; 6.05 | 0.095; 6.04 | 0.082; 6.38 | **0.078; 3.65** | **0.065; 3.92** | 0.063; 3.67 |
| EM | VS | 0.108; 5.53 | 0.088; 4.82 | 0.076; 4.98 | **0.077; 3.26** | **0.062; 3.02** | **0.059; 3.03** |
| BBSL-hard | None | 0.227; 12.49 | 0.19; 13.67 | 0.154; 13.18 | 0.196; 13.12 | 0.157; 13.15 | 0.128; 12.18 |
| BBSL-soft | None | 0.17; 10.69 | 0.127; 10.67 | 0.105; 9.7 | 0.131; 11.43 | 0.096; 10.27 | 0.091; 9.65 |
| BBSL-soft | TS | 0.16; 10.23 | 0.119; 9.55 | 0.1; 9.11 | 0.122; 10.84 | 0.092; 9.42 | 0.089; 9.29 |
| BBSL-soft | NBVS | 0.176; 11.21 | 0.121; 9.36 | 0.106; 10.22 | 0.125; 9.81 | 0.093; 9.06 | 0.086; 8.41 |
| BBSL-soft | BCTS | 0.167; 10.34 | 0.116; 8.9 | 0.096; 8.85 | 0.107; 8.18 | 0.079; 6.29 | 0.077; 6.54 |
| BBSL-soft | VS | 0.159; 10.13 | 0.118; 9.3 | 0.093; 8.25 | 0.101; 7.29 | 0.079; 6.59 | 0.077; 6.68 |
| RLLS-hard | None | 0.148; 9.41 | 0.161; 12.31 | 0.143; 12.16 | 0.104; 8.47 | 0.089; 8.8 | 0.087; 8.31 |
| RLLS-soft | None | 0.134; 8.22 | 0.123; 9.88 | 0.102; 9.05 | 0.092; 6.6 | 0.082; 7.27 | 0.081; 7.03 |
| RLLS-soft | TS | 0.133; 8.12 | 0.116; 8.64 | 0.098; 8.35 | 0.092; 6.6 | 0.083; 7.56 | 0.083; 7.83 |
| RLLS-soft | NBVS | 0.124; 7.69 | 0.105; 7.36 | 0.101; 9.48 | 0.092; 6.32 | 0.086; 8.3 | 0.085; 8.36 |
| RLLS-soft | BCTS | 0.127; 7.69 | 0.11; 7.75 | 0.094; 8.14 | 0.089; 5.47 | 0.078; 6.28 | 0.078; 6.56 |
| RLLS-soft | VS | 0.124; 7.41 | 0.11; 7.9 | 0.091; 7.28 | 0.089; 5.77 | 0.079; 6.37 | 0.077; 6.52 |

Table H.2: **Kaggle Diabetic Retinopathy Detection: comparison of all calibration and domain adaptation methods, using MSE (Sec. 3.3) as the metric.** Like **Table H.1**, but with JS Divergence (**Sec. B**) as the metric rather than MSE.

| Shift Estimator | Calibration Method | $\rho = 0.5$ | | | $\rho = 0.9$ | | |
|---|---|---|---|---|---|---|---|
| | | n=500 | n=1000 | n=1500 | n=500 | n=1000 | n=1500 |
| EM | None | **0.077; 0.36** | **0.059; 0.15** | **0.054; 0.26** | 0.111; 1.99 | 0.1; 2.15 | 0.102; 2.44 |
| BBSL-hard | None | 0.227; 3.2 | 0.19; 3.39 | 0.154; 3.25 | 0.196; 3.14 | 0.157; 3.16 | 0.128; 2.86 |
| BBSL-soft | None | 0.17; 2.54 | 0.127; 1.98 | 0.105; 1.97 | 0.131; 2.35 | 0.096; 1.84 | 0.091; 1.79 |
| RLLS-hard | None | 0.148; 2.16 | 0.161; 2.73 | 0.143; 2.77 | 0.104; 1.61 | **0.089; 1.63** | **0.087; 1.6** |
| RLLS-soft | None | 0.134; 1.74 | 0.123; 1.75 | 0.102; 1.75 | **0.092; 0.91** | **0.082; 1.22** | **0.081; 1.31** |
| EM | TS | **0.09; 0.26** | **0.068; 0.2** | **0.061; 0.15** | 0.104; 1.04 | 0.094; 1.05 | 0.094; 1.16 |
| BBSL-soft | TS | 0.16; 1.65 | 0.119; 1.59 | 0.1; 1.54 | 0.122; 1.4 | 0.092; 1.17 | **0.089; 1.06** |
| RLLS-soft | TS | 0.133; 1.09 | 0.116; 1.21 | 0.098; 1.31 | **0.092; 0.56** | **0.083; 0.78** | **0.083; 0.78** |
| EM | NBVS | **0.107; 0.42** | **0.089; 0.52** | **0.079; 0.42** | 0.11; 1.2 | 0.1; 1.21 | 0.102; 1.31 |
| BBSL-soft | NBVS | 0.176; 1.6 | 0.121; 1.48 | 0.106; 1.42 | 0.125; 1.2 | 0.093; 1.0 | **0.086; 0.89** |
| RLLS-soft | NBVS | 0.124; 0.98 | 0.105; 1.0 | 0.101; 1.16 | **0.092; 0.6** | **0.086; 0.79** | **0.085; 0.8** |
| EM | BCTS | **0.111; 0.6** | **0.095; 0.62** | **0.082; 0.72** | **0.078; 0.61** | **0.065; 0.65** | **0.063; 0.54** |
| BBSL-soft | BCTS | 0.167; 1.49 | 0.116; 1.37 | 0.096; 1.31 | 0.107; 1.42 | 0.079; 1.18 | 0.077; 1.21 |
| RLLS-soft | BCTS | 0.127; 0.91 | 0.11; 1.01 | 0.094; 0.97 | 0.089; 0.97 | 0.078; 1.17 | 0.078; 1.25 |
| EM | VS | **0.108; 0.55** | **0.088; 0.5** | **0.076; 0.61** | **0.077; 0.51** | **0.062; 0.47** | **0.059; 0.55** |
| BBSL-soft | VS | 0.159; 1.6 | 0.118; 1.43 | 0.093; 1.4 | 0.101; 1.46 | 0.079; 1.31 | 0.077; 1.24 |
| RLLS-soft | VS | 0.124; 0.85 | 0.11; 1.07 | 0.091; 0.99 | 0.089; 1.03 | 0.079; 1.22 | 0.077; 1.21 |

Table H.3: **Kaggle Diabetic Retinopathy Detection: Comparison of EM, BBSL and RLLS.** Analogous to **Table 4**, but with JS Divergence (**Sec. B**) as the metric rather than MSE.

## I NLL CORRESPONDS BETTER TO BENEFITS IN LABEL SHIFT ADAPTATION

To investigate whether NLL or ECE corresponded better to the benefits offered by a calibration method in the context of label shift adaptation, we adopted the following strategy: in a given experimental run, we identified the calibration method that provided the best NLL (or ECE) on the unshifted test set. We then looked at the performance of label shift adaptation using this calibration method. Note that the calibration method selected can differ from one run to the next. Across datasets, we observed that, by and large, selecting a calibration method according to the NLL produced better performance after domain adaptation as compared to selecting a calibration method according to ECE. Results are show in the tables below.

| Shift Estimator | Calibration Method | $\alpha = 0.1$ | | | $\alpha = 1.0$ | | | $\alpha = 10$ | | |
|---|---|---|---|---|---|---|---|---|---|---|
| | | n=2000 | n=4000 | n=8000 | n=2000 | n=4000 | n=8000 | n=2000 | n=4000 | n=8000 |
| EM | Best NLL | 7.332; 0.3 | 7.326; 0.32 | **7.37; 0.28** | 2.593; 0.36 | **2.664; 0.09** | **2.688; 0.06** | 0.764; 0.42 | **0.839; 0.04** | **0.884; 0.03** |
| EM | Best ECE | 7.298; 0.7 | 7.302; 0.68 | 7.318; 0.72 | 2.548; 0.64 | 2.172; 0.91 | 2.204; 0.94 | 0.741; 0.58 | 0.225; 0.96 | 0.276; 0.97 |

Table I.1: **CIFAR10: NLL vs ECE, $\Delta$%Accuracy, dirichlet shift.** Entry in "calibration method" column indicates how the calibration method for any given run was selected: either according to whether it produced the best NLL or whether it produced the best ECE, where NLL and ECE were calculated on the unshifted test set. Value before the semicolon is the average change in %accuracy relative to unadapted predictions. Value after the semicolon is the average rank of the given metric relative to the other metric in the pair. A bold value is significantly better than the non-bold value in the pair using a paired Wilcoxon test at $p \leq 0.01$. See **Sec. 4.1** for details on the experimental setup.

| Shift Estimator | Calibration Method | $\rho = 0.01$ | | | $\rho = 0.9$ | | |
|---|---|---|---|---|---|---|---|
| | | n=2000 | n=4000 | n=8000 | n=2000 | n=4000 | n=8000 |
| EM | Best NLL | **1.192; 0.17** | **1.253; 0.21** | **1.301; 0.15** | 17.724; 0.47 | **17.779; 0.08** | **17.84; 0.07** |
| EM | Best ECE | 1.053; 0.83 | 1.149; 0.79 | 1.16; 0.85 | 17.727; 0.53 | 17.26; 0.92 | 17.288; 0.93 |

Table I.2: **CIFAR10: NLL vs. ECE, metric: $\Delta$%accuracy, "tweak-one" shift**. Analogous to **Table I.1**. The "tweak-one" shift strategy is explained in **Sec. 4.1**.

| Shift Estimator | Calibration Method | α = 0.1 | | | α = 1.0 | | | α = 10 | | |
|---|---|---|---|---|---|---|---|---|---|---|
| | | n=2000 | n=4000 | n=8000 | n=2000 | n=4000 | n=8000 | n=2000 | n=4000 | n=8000 |
| EM | Best NLL | 2.003; 0.33 | 1.625; 0.4 | **1.488; 0.36** | 1.6; 0.38 | **1.24; 0.14** | 0.918; 0.12 | 1.518; 0.51 | **1.041; 0.31** | 0.775; 0.26 |
| EM | Best ECE | 2.144; 0.67 | 1.635; 0.6 | 1.642; 0.64 | 1.647; 0.62 | 1.714; 0.86 | 1.398; 0.88 | 1.515; 0.49 | 1.156; 0.69 | 0.882; 0.74 |
| BBSL-soft | Best NLL | 6.636; 0.41 | 5.493; 0.35 | 4.689; 0.49 | 2.543; 0.49 | 1.804; 0.47 | 1.356; 0.55 | 1.559; 0.5 | 1.086; 0.41 | 0.789; 0.38 |
| BBSL-soft | Best ECE | 6.65; 0.59 | 5.526; 0.65 | 4.703; 0.51 | 2.55; 0.51 | 1.87; 0.53 | 1.339; 0.45 | 1.559; 0.5 | 1.124; 0.59 | 0.818; 0.62 |
| RLLS-soft | Best NLL | 6.523; 0.36 | 5.399; 0.36 | 4.625; 0.49 | 2.545; 0.49 | 1.805; 0.47 | 1.356; 0.55 | 1.559; 0.5 | 1.086; 0.41 | 0.789; 0.38 |
| RLLS-soft | Best ECE | 6.55; 0.64 | 5.431; 0.64 | 4.638; 0.51 | 2.552; 0.51 | 1.87; 0.53 | 1.339; 0.45 | 1.559; 0.5 | 1.124; 0.59 | 0.818; 0.62 |

Table I.3: **CIFAR10: NLL vs. ECE, metric: JS Divergence, dirichlet shift**. Analogous to **Table I.1**, but using JS Divergence (**Sec. B**) as the metric rather than change in %accuracy.

| Shift Estimator | Calibration Method | ρ = 0.01 | | | ρ = 0.9 | | |
|---|---|---|---|---|---|---|---|
| | | n=2000 | n=4000 | n=8000 | n=2000 | n=4000 | n=8000 |
| EM | Best NLL | **1.566; 0.29** | **1.0; 0.33** | **0.793; 0.29** | 2.787; 0.39 | **2.339; 0.13** | **2.132; 0.04** |
| EM | Best ECE | 1.695; 0.71 | 1.099; 0.67 | 0.906; 0.71 | 2.908; 0.61 | 3.388; 0.87 | 3.223; 0.96 |
| BBSL-soft | Best NLL | 2.012; 0.4 | 1.409; 0.51 | 0.985; 0.55 | 7.431; 0.59 | **5.737; 0.25** | **3.84; 0.16** |
| BBSL-soft | Best ECE | 2.005; 0.6 | **1.389; 0.49** | **0.959; 0.45** | 7.393; 0.41 | 6.214; 0.75 | 4.328; 0.84 |
| RLLS-soft | Best NLL | 2.011; 0.4 | 1.409; 0.51 | 0.985; 0.55 | 7.433; 0.58 | **5.738; 0.25** | **3.841; 0.16** |
| RLLS-soft | Best ECE | 2.004; 0.6 | **1.389; 0.49** | **0.959; 0.45** | 7.395; 0.42 | 6.214; 0.75 | 4.328; 0.84 |

Table I.4: **CIFAR10: NLL vs. ECE, metric: JS Divergence, "tweak-one" shift**. Analogous to **Table I.1**.

| Shift Estimator | Calibration Method | α = 0.1 | | | α = 1.0 | | | α = 10 | | |
|---|---|---|---|---|---|---|---|---|---|---|
| | | n=2000 | n=4000 | n=8000 | n=2000 | n=4000 | n=8000 | n=2000 | n=4000 | n=8000 |
| EM | Best NLL | **0.17093; 0.17** | **0.07813; 0.27** | **0.05336; 0.27** | 0.1631; 0.44 | **0.09884; 0.08** | **0.05226; 0.08** | 0.19969; 0.5 | **0.09141; 0.24** | **0.04949; 0.22** |
| EM | Best ECE | 1.58795; 0.83 | 0.85381; 0.73 | 0.48624; 0.73 | 0.16126; 0.56 | 0.48293; 0.92 | 0.38694; 0.92 | 0.19985; 0.5 | 0.11969; 0.76 | 0.06922; 0.78 |
| BBSL-soft | Best NLL | 0.80638; 0.43 | 0.28839; 0.4 | 0.18902; 0.47 | 0.27634; 0.57 | 0.14499; 0.5 | 0.07681; 0.46 | 0.21012; 0.51 | **0.09854; 0.41** | **0.05166; 0.4** |
| BBSL-soft | Best ECE | 0.82049; 0.57 | 0.30148; 0.6 | 0.18619; 0.53 | 0.27356; 0.43 | 0.15819; 0.5 | 0.07883; 0.54 | 0.2095; 0.49 | 0.10475; 0.59 | 0.05539; 0.6 |
| RLLS-soft | Best NLL | 0.71635; 0.41 | 0.28071; 0.38 | 0.18712; 0.49 | 0.27385; 0.57 | 0.14469; 0.5 | 0.07673; 0.46 | 0.21012; 0.51 | **0.09854; 0.41** | **0.05166; 0.4** |
| RLLS-soft | Best ECE | 0.70332; 0.59 | 0.28998; 0.62 | 0.18421; 0.51 | 0.27112; 0.43 | 0.158; 0.5 | 0.0788; 0.54 | 0.2095; 0.49 | 0.10475; 0.59 | 0.05539; 0.6 |

Table I.5: **CIFAR10: NLL vs. ECE, metric: MSE, dirichlet shift**. Analogous to **Table I.1**, but using MSE (**Sec. 3.3**) as the metric rather than change in %accuracy.

| Shift Estimator | Calibration Method | ρ = 0.01 | | | ρ = 0.9 | | |
|---|---|---|---|---|---|---|---|
| | | n=2000 | n=4000 | n=8000 | n=2000 | n=4000 | n=8000 |
| EM | Best NLL | **0.14964; 0.27** | **0.06447; 0.3** | **0.03926; 0.35** | 0.2213; 0.43 | **0.13654; 0.02** | **0.1036; 0.02** |
| EM | Best ECE | 0.17892; 0.73 | 0.07276; 0.7 | 0.04478; 0.65 | 0.26599; 0.57 | 10.91926; 0.98 | 10.94244; 0.98 |
| BBSL-soft | Best NLL | 0.17484; 0.44 | 0.08129; 0.52 | 0.04739; 0.57 | 1.80777; 0.54 | **0.93671; 0.24** | **0.50136; 0.14** |
| BBSL-soft | Best ECE | 0.17423; 0.56 | **0.0799; 0.48** | **0.04615; 0.43** | 1.79819; 0.46 | 1.1714; 0.76 | 0.67683; 0.86 |
| RLLS-soft | Best NLL | 0.17377; 0.44 | 0.08129; 0.52 | 0.04739; 0.57 | 1.76868; 0.54 | **0.92956; 0.23** | **0.49941; 0.13** |
| RLLS-soft | Best ECE | 0.17305; 0.56 | **0.0799; 0.48** | **0.04615; 0.43** | 1.75943; 0.46 | 1.16983; 0.77 | 0.6753; 0.87 |

Table I.6: **CIFAR10: NLL vs ECE, metric: MSE, "tweak-one" shift.** Analogous to **Table I.1**.

| Shift Estimator | Calibration Method | α = 0.1 | | | α = 1.0 | | | α = 10.0 | | |
|---|---|---|---|---|---|---|---|---|---|---|
| | | n=7000 | n=8500 | n=10000 | n=7000 | n=8500 | n=10000 | n=7000 | n=8500 | n=10000 |
| EM | Best NLL | **26.889; 0.3** | **26.901; 0.31** | **26.954; 0.31** | 21.94; 0.28 | **22.097; 0.28** | 22.183; 0.2 | **21.201; 0.22** | 21.41; 0.21 | 21.36; 0.2 |
| EM | Best ECE | 26.332; 0.7 | 26.323; 0.69 | 26.464; 0.69 | 21.588; 0.72 | 21.711; 0.72 | 21.708; 0.8 | 20.933; 0.78 | 21.103; 0.79 | 21.09; 0.8 |

Table I.7: **CIFAR100: NLL vs ECE, metric: Δ%Accuracy, dirichlet shift.** Analogous to **Table I.1**

| Shift Estimator | Calibration Method | α = 0.1 | | | α = 1.0 | | | α = 10.0 | | |
|---|---|---|---|---|---|---|---|---|---|---|
| | | n=7000 | n=8500 | n=10000 | n=7000 | n=8500 | n=10000 | n=7000 | n=8500 | n=10000 |
| EM | Best NLL | **0.113; 0.31** | **0.112; 0.27** | **0.112; 0.29** | **0.111; 0.17** | **0.108; 0.09** | **0.107; 0.1** | **0.108; 0.03** | **0.106; 0.03** | **0.104; 0.0** |
| EM | Best ECE | 0.119; 0.69 | 0.119; 0.73 | 0.119; 0.71 | 0.118; 0.83 | 0.117; 0.91 | 0.116; 0.9 | 0.116; 0.97 | 0.114; 0.97 | 0.112; 1.0 |
| BBSL-soft | Best NLL | 0.226; 0.41 | 0.22; 0.35 | **0.215; 0.39** | **0.188; 0.3** | 0.182; 0.31 | **0.177; 0.24** | **0.177; 0.2** | 0.171; 0.2 | **0.165; 0.29** |
| BBSL-soft | Best ECE | 0.226; 0.59 | 0.22; 0.65 | 0.216; 0.61 | 0.189; 0.7 | 0.183; 0.69 | 0.178; 0.76 | 0.178; 0.8 | 0.172; 0.8 | 0.166; 0.71 |
| RLLS-soft | Best NLL | 0.186; 0.5 | 0.179; 0.42 | 0.176; 0.48 | 0.155; 0.49 | 0.149; 0.48 | 0.144; 0.51 | **0.148; 0.34** | 0.142; 0.46 | 0.137; 0.49 |
| RLLS-soft | Best ECE | 0.186; 0.5 | 0.179; 0.58 | 0.176; 0.52 | 0.155; 0.51 | 0.149; 0.52 | 0.144; 0.49 | 0.149; 0.66 | 0.142; 0.54 | 0.137; 0.51 |

Table I.8: **CIFAR100: NLL vs ECE, metric: JS Divergence, dirichlet shift.** Analogous to **Table I.1**

| Shift Estimator | Calibration Method | α = 0.1 | | | α = 1.0 | | | α = 10.0 | | |
|---|---|---|---|---|---|---|---|---|---|---|
| | | n=7000 | n=8500 | n=10000 | n=7000 | n=8500 | n=10000 | n=7000 | n=8500 | n=10000 |
| EM | Best NLL | **0.1994; 0.37** | **0.2011; 0.36** | **0.20436; 0.35** | **0.13788; 0.22** | **0.1307; 0.23** | **0.12736; 0.26** | **0.10309; 0.2** | **0.09864; 0.2** | **0.09667; 0.17** |
| EM | Best ECE | 0.28904; 0.63 | 0.27676; 0.64 | 0.26944; 0.65 | 0.15848; 0.78 | 0.14828; 0.77 | 0.14304; 0.74 | 0.11248; 0.8 | 0.10512; 0.8 | 0.10192; 0.83 |
| BBSL-soft | Best NLL | **0.94791; 0.36** | **0.66421; 0.36** | **0.57766; 0.37** | **0.23665; 0.24** | **0.18917; 0.23** | **0.16374; 0.2** | **0.15332; 0.24** | **0.11667; 0.23** | **0.09866; 0.1** |
| BBSL-soft | Best ECE | 1.01696; 0.64 | 0.69643; 0.64 | 0.60503; 0.63 | 0.24203; 0.76 | 0.19391; 0.77 | 0.16837; 0.8 | 0.1567; 0.76 | 0.11969; 0.77 | 0.10204; 0.9 |
| RLLS-soft | Best NLL | 0.64403; 0.5 | 0.52134; 0.54 | 0.47947; 0.54 | 0.1941; 0.48 | 0.15799; 0.55 | 0.1352; 0.43 | 0.11958; 0.39 | 0.0966; 0.45 | **0.08386; 0.27** |
| RLLS-soft | Best ECE | 0.65047; 0.5 | 0.52242; 0.46 | 0.48347; 0.46 | 0.19225; 0.52 | 0.15747; 0.45 | 0.13543; 0.57 | 0.12059; 0.61 | 0.09732; 0.55 | 0.08476; 0.73 |

Table I.9: **CIFAR100: NLL vs ECE, metric: MSE, dirichlet shift.** Analogous to **Table I.1**

| Shift Estimator | Calibration Method | $\rho = 0.5$ | | | $\rho = 0.9$ | | |
|---|---|---|---|---|---|---|---|
| | | $n$=500 | $n$=1000 | $n$=1500 | $n$=500 | $n$=1000 | $n$=1500 |
| EM | Best NLL | **3.79; 0.21** | **4.315; 0.26** | **4.543; 0.19** | **3.548; 0.02** | 3.57; 0.0 | **3.746; 0.02** |
| EM | Best ECE | 3.49; 0.79 | 4.099; 0.74 | 4.179; 0.81 | 2.074; 0.98 | 3.57; 1.0 | 2.405; 0.98 |

Table I.10: **KaggleDR: NLL vs ECE, metric: $\Delta$%Accuracy.** Shift strategy modifies the proportion of healthy examples. Analogous to **Table I.1**

| Shift Estimator | Calibration Method | $\rho = 0.5$ | | | $\rho = 0.9$ | | |
|---|---|---|---|---|---|---|---|
| | | $n$=500 | $n$=1000 | $n$=1500 | $n$=500 | $n$=1000 | $n$=1500 |
| EM | Best NLL | 0.11; 0.42 | 0.093; 0.31 | 0.079; 0.33 | **0.078; 0.08** | 0.062; 0.0 | **0.059; 0.07** |
| EM | Best ECE | 0.104; 0.58 | 0.092; 0.69 | 0.079; 0.67 | 0.11; 0.92 | 0.062; 1.0 | 0.102; 0.93 |
| BBSL-soft | Best NLL | 0.166; 0.37 | 0.12; 0.32 | **0.096; 0.31** | **0.107; 0.24** | 0.079; 0.0 | **0.077; 0.32** |
| BBSL-soft | Best ECE | 0.158; 0.63 | 0.123; 0.68 | 0.101; 0.69 | 0.125; 0.76 | 0.079; 1.0 | 0.086; 0.68 |
| RLLS-soft | Best NLL | 0.128; 0.44 | 0.112; 0.38 | **0.093; 0.31** | **0.089; 0.38** | 0.079; 0.0 | **0.077; 0.38** |
| RLLS-soft | Best ECE | **0.123; 0.56** | 0.109; 0.62 | 0.098; 0.69 | 0.092; 0.62 | 0.079; 1.0 | 0.085; 0.62 |

Table I.11: **KaggleDR: NLL vs ECE, metric: JS Divergence.** Shift strategy modifies the proportion of healthy examples. Analogous to **Table I.1**

| Shift Estimator | Calibration Method | $\rho = 0.5$ | | | $\rho = 0.9$ | | |
|---|---|---|---|---|---|---|---|
| | | $n$=500 | $n$=1000 | $n$=1500 | $n$=500 | $n$=1000 | $n$=1500 |
| EM | Best NLL | 1.076; 0.3 | **0.46; 0.24** | 0.319; 0.32 | **0.069; 0.07** | 0.032; 0.0 | **0.029; 0.01** |
| EM | Best ECE | 1.028; 0.7 | 0.549; 0.76 | 0.354; 0.68 | 0.168; 0.93 | 0.032; 1.0 | 0.125; 0.99 |
| BBSL-soft | Best NLL | 61.132; 0.43 | **1.439; 0.27** | 0.875; 0.29 | **0.747; 0.27** | 0.049; 0.0 | **0.042; 0.34** |
| BBSL-soft | Best ECE | **8.74; 0.57** | 2.181; 0.73 | 0.932; 0.71 | 2.678; 0.73 | 0.049; 1.0 | 0.067; 0.66 |
| RLLS-soft | Best NLL | 2.445; 0.44 | 0.859; 0.33 | 0.726; 0.31 | 0.066; 0.38 | 0.042; 0.0 | **0.035; 0.43** |
| RLLS-soft | Best ECE | **2.089; 0.56** | 0.867; 0.67 | 0.742; 0.69 | 0.072; 0.62 | 0.042; 1.0 | 0.046; 0.57 |

Table I.12: **KaggleDR: NLL vs ECE, metric: MSE.** Shift strategy modifies the proportion of healthy examples. Analogous to **Table I.1**

