# OpenReview forum: "Adapting to Label Shift with Bias-Corrected Calibration"
_ICLR.cc/2020/Conference — Reject_

### Official Review · AnonReviewer3 · 2019-10-22
**Official Blind Review #3**

**Rating:** 1

**Review:**

The paper considers the label shift problem and investigates the role of the calibration in improving the results of two different domain adaptation solutions including EM and BBSE. They show with having better estimation of conditional distribution in the source domain the final label distribution obtained by EM and BBSE in the target domain is more accurate. They conduct the experiments on three different datasets i.e. CIFAR10, CIFAR100 and Kaggel Diabetic Retinopathy as the proof of concept.

Overall, I think the paper should be rejected as it suffers from not high enough level of novelty in proposed method. The paper used different variants of famous Platt Scaling family approaches as the calibration methods to improve the estimation of conditional distribution of the training data $p(y|x)$ and showed the positive influence of that on label shift approaches like EM and BBSE, which is not enough contribution.

The paper is well-written and point the interesting problem. But the way of reporting the results are not clear enough. It would be better that the accuracy is reported for the baseline, EM,  BBSE-Soft and BBSE-hard at the same table with mean and std values. The tables some how should show the impact of using the calibration to improve significantly the final label shift accuracy. But in this way of reporting the results it is not clear how big calibration can improve the final accuracy. Comparing to other baselines like RLLS [1] is also missing in this table.

Pointing out that ECE is not a good metric to measure the calibration is also reported before in the related research works [2].


References
[1] Azizzadenesheli, Kamyar, et al. "Regularized learning for domain adaptation under label shifts." arXiv preprint arXiv:1903.09734 (2019).
[2] Vaicenavicius, Juozas, et al. "Evaluating model calibration in classification." arXiv preprint arXiv:1902.06977 (2019).


**Experience Assessment:**

I do not know much about this area.

**Review Assessment: Checking Correctness Of Derivations And Theory:**

I did not assess the derivations or theory.

**Review Assessment: Checking Correctness Of Experiments:**

I assessed the sensibility of the experiments.

**Review Assessment: Thoroughness In Paper Reading:**

I read the paper at least twice and used my best judgement in assessing the paper.

---

> ### Author Response · Authors · 2019-11-10
> **response to review #3**
>
> We thank the reviewer for their feedback and have revised the manuscript to clarify the contributions. We would specifically like to highlight the following points:
>
> Having included Regularized Learning under Label Shift in the comparisons (Azizzadenesheli et al.) in the comparisons, we find that our proposed approach of  Expectation Maximization combined with bias-corrected calibration achieves state-of-the-art results in our experiments, outperforming both RLLS and BBSL on a significant majority of comparisons. Note that the RLLS and BBSL papers did not include comparisons to the EM approach - in fact, benchmarks against the EM algorithm have been missing from multiple recent high-profile works [1, 2, 3], seemingly due to a misconception regarding the scalability of the algorithm.
>
> The specific calibration approach that we recommend in order to achieve state-of-the-art results is an approach that is not the dominant approach recommended in the literature. In particular, we find that it is important to use a calibration approach that contains class-specific bias parameters that are capable of correcting for systematic bias in the calibrated probabilities. Although one such approach - Vector Scaling - was introduced in Guo et al., it was not the method that the authors ultimately recommended for calibration because it achieved slightly worse ECE compared to Temperature Scaling. As a result, Temperature Scaling - which does not contain class-specific parameters - has become the dominant approach in the literature. An important contribution of our paper is to show that Temperature Scaling is not the best choice for domain adaptation to label shift.
>
> We make two contributions to the EM algorithm itself. The first is a theoretically principled strategy for computing the source-domain priors that substantially improves the robustness of the algorithm in situations where the predicted probabilities retain systematic bias. The second, requested by Reviewer 2, is a proof showing that maximizing the likelihood is a convex optimization problem - thus, we can expect EM will converge to a global optimum.
>
> We hope you will revisit this submission with the above considerations in mind.
>
>
> [1] Zhang, Kun, Bernhard Schölkopf, Krikamol Muandet, and Zhikun Wang. "Domain adaptation under target and conditional shift." In International Conference on Machine Learning, pp. 819-827. 2013.
>
> [2] Lipton, Zachary C., Yu-Xiang Wang, and Alex Smola. "Detecting and correcting for label shift with black box predictors." arXiv preprint arXiv:1802.03916 (2018).
>
> [3] Azizzadenesheli, Kamyar, Anqi Liu, Fanny Yang, and Animashree Anandkumar. "Regularized learning for domain adaptation under label shifts." arXiv preprint arXiv:1903.09734 (2019).

---

### Official Review · AnonReviewer1 · 2019-10-23
**Official Blind Review #1**

**Rating:** 3

**Review:**

This work conducted exhaustive experiments for a label shift problem with EM and BBSE over CIFAR10/100 and retinopathy detection datasets. In addition, as for model calibration, they also considered temperature and vector scaling and introduced intermediaries between those, namely, NBVS and TBVS.

Overall, the paper is well organized and provides rigorous experiments with conclusions based on the empirical observations. As stated in the paper, the main contribution of this work is to explore the impact of calibration.

However, it is still doubtful whether those empirical results can be generalized as there is no analytical and thoughtful discussions. Further, the label shift was simulated by means of dirichlet shift on both datasets. It would be great to apply the method on real cases.

**Experience Assessment:**

I do not know much about this area.

**Review Assessment: Checking Correctness Of Derivations And Theory:**

N/A

**Review Assessment: Checking Correctness Of Experiments:**

I assessed the sensibility of the experiments.

**Review Assessment: Thoroughness In Paper Reading:**

I read the paper at least twice and used my best judgement in assessing the paper.

---

> ### Author Response · Authors · 2019-11-10
> **response to review #1**
>
> We thank the reviewer for the feedback and have revised the manuscript to clarify the contributions. We would like to highlight the following points:
>
> While generalization bounds are beyond the scope of this paper, our work revisits EM as a viable yet missing method from the recent literature ([1], [2], [3]). The EM algorithm comes with strong theoretical grounding when the probabilities are calibrated [4]. However, at the time when the EM algorithm was introduced for label shift adaptation, research on calibration techniques that are specifically tailored to DNNs had not been conducted.
>
> Our contribution is to unify the two lines of research - calibration techniques and domain adaptation - to show that one can achieve strong performance by using calibration in conjunction with the EM algorithm. A key observation that we made is that the popular calibration approach of Temperature Scaling can be improved upon by the addition of class-specific bias parameters. Vector Scaling, which does extend Temperature Scaling by including class-specific bias parameters, was not recommended in the paper that introduced it because it was observed to produce worse performance on the ECE metric, and thus has not been adopted. Our work demonstrates the importance of retaining these class-specific bias parameters in the context of label shift. Until we had investigated the importance of including bias correction, we ourselves were adopting the Temperature Scaling approach.
>
> When we proposed an alternative method for calculating source-domain priors for EM in the presence of poor calibration, we provided the appropriate analytical discussion (lemma 1, section 3.2). In addition, as requested by Reviewer 2, we have revised the manuscript to include a proof (appendix A) showing that maximizing the likelihood is a convex optimization problem - thus, we can expect EM will converge to a global optimum. Empirically, we verified that this technique enables EM to perform very well at shift estimation across different datasets compared to the recently-introduced Black-Box Shift Estimation, even when the probabilities are not well calibrated. More research into why EM is so effective would be the subject of future work.
>
> Having included Regularized Learning under Label Shift in the comparisons (Azizzadenesheli et al.) in the revised comparisons, we find that our proposed approach of EM combined with bias-corrected calibration now achieves state-of-the-art results in our experiments, outperforming both RLLS and BBSL on a significant majority of comparisons. Note that the RLLS and BBSL papers did not include comparisons to the EM approach - in fact, benchmarks against the EM algorithm have been missing from multiple recent high-profile works [1, 2, 3], seemingly due to a misconception regarding the scalability of the algorithm.
>
> Regarding the distribution shifts, Dirichlet and tweak one shifts are standard in the literature and capture the first order effects of label shifts for several real world applications ([2], [3]). We also revised the manuscript to include comparisons against RLLS ([3]) on all the datasets and included new comparisons on MNIST.
>
> We believe this work is rigorous and experimentally sound. We hope you reconsider this submission in light of the above comments.
>
>
> [1] Zhang, Kun, Bernhard Schölkopf, Krikamol Muandet, and Zhikun Wang. "Domain adaptation under target and conditional shift." In International Conference on Machine Learning, pp. 819-827. 2013.
>
> [2] Lipton, Zachary C., Yu-Xiang Wang, and Alex Smola. "Detecting and correcting for label shift with black box predictors." arXiv preprint arXiv:1802.03916 (2018).
>
> [3] Azizzadenesheli, Kamyar, Anqi Liu, Fanny Yang, and Animashree Anandkumar. "Regularized learning for domain adaptation under label shifts." arXiv preprint arXiv:1903.09734 (2019).
>
> [4] Saerens, Marco, Patrice Latinne, and Christine Decaestecker. "Adjusting the outputs of a classifier to new a priori probabilities: a simple procedure." Neural computation 14, no. 1 (2002): 21-41.

---

### Official Review · AnonReviewer2 · 2019-10-26
**Official Blind Review #2**

**Rating:** 6

**Review:**

This paper builds upon recent work on detecting and correcting for label shift.
They explore both the BBSE algorithm analyzed in Detecting and Correcting for Label Shift (2018)
and another approach based on EM where the predictive posteriors and test set label distributions
are iteratively computed, each an update based on the estimate of the other.

Crucially, while the former method requires only that the confusion matrix be invertible,
the  latter method only appears valid under strong assumptions including the calibration fo the classifier.
Thus the authors propose an approach for “bias-corrected calibration”
and shows that bias-corrected calibration can improve the performance of BBSE and EM.
The method is crucial for EM and with it, the results seem to show that EM,
in the large sample (8000 examples) regime and with good initial classifiers
 (on the relatively easy CIFAR10 task with a strong baseline)
that EM outperforms BBSE.

The paper is easy to follow an the authors should also be credited for releasing
code anonymously with which we could reproduce their results.

I have as few specific concerns/questions about the paper that I would like the authors to address:

 * They consider JS divergence as a metric for evaluation. But they don’t consider other metrics
  like the error in weight estimates which is considered in most of the prior work
 * They don’t compare their results with regularizations suggested on top of BBSE, particularly Azizzadenesheli et. al. https://arxiv.org/abs/1903.09734.
 * They compare methods for particularly limited ranges of Dirichlet shift (\alpha=0.1,1.0).
  * What happens when the \alpha increases to have less severe shifts?
Optimizing ELBO with EM can lead to local convergence to the likelihood function when the likelihood is not unimodal.
 *  Is this likelihood function unimodal?  Does the EM approach converges to MLE under some appropriate initialization and assumptions?

A small presentation note: many of the papers are reporting the same metric and ought to be grouped as a large table, not as many tables. Also every table should state clearly what it is reporting in the caption, not just referring to earlier tables.

=======Update
I have read the rebuttal and appreciate that the authors took the time to establish the concavity of the likelihood function for EM. Overall this paper makes an interesting contribution in establishing the usefulness of the likelihood formulation (here optimized by EM) of label shift estimation and its apparent benefits over BBSE in some settings. I am happy to keep my score despite apparent disagreement from the other reviewers. I must say that some other reviews were disappointingly lacking in thoroughness.

The paper still leaves open some serious questions, e.g. --- why is this bias correction heuristic so effective vis-a-vis EM and is this explained by its performance at the calibration task itself? The original temperature scaling paper reported a similar heuristic yet didn't see such a benefit wrt their metrics. Why is it so useful here?

Still, while this paper can be improved in some key ways, it does make an interesting contribution.

**Experience Assessment:**

I have published in this field for several years.

**Review Assessment: Checking Correctness Of Derivations And Theory:**

I carefully checked the derivations and theory.

**Review Assessment: Checking Correctness Of Experiments:**

I carefully checked the experiments.

**Review Assessment: Thoroughness In Paper Reading:**

I read the paper thoroughly.

---

> ### Author Response · Authors · 2019-11-10
> **response to review #2**
>
> We thank the reviewer for their excellent suggestions and have incorporated them in the revision. Specifically:
>
> “They consider JS divergence as a metric for evaluation. But they don’t consider other metrics like the error in weight estimates which is considered in most of the prior work” + “They don’t compare their results with regularizations suggested on top of BBSE, particularly Azizzadenesheli et al.”
> We have included Regularized Learning under Label Shift (RLLS) by Azizzadenesheli et al. and have added the mean squared error in weight estimates as a metric. We find that our overall conclusions are strengthened: when the predictions are not well calibrated, EM produces mixed results and RLLS has a tendency to achieve the best results. However, when Bias-Corrected Temperature Scaling or Vector Scaling are used for calibration, EM tends to outperform both BBSL and RLLS (for consistency with the prior literature, we have replaced our references to “BBSE” with “BBSL” in the manuscript). Because BBSL and RLLS are considered the state-of-the-art among computationally scalable methods for label shift estimation, we can now state that our proposed hybrid approach of EM + bias-corrected calibration achieves state-of-the-art results in our experiments.
>
> “They compare methods for particularly limited ranges of Dirichlet shift (\alpha=0.1, 1.0). What happens when the \alpha increases to have less severe shifts?”
> We have added \alpha=10 to the tables for CIFAR10 and CIFAR100 in the main text. We find that the overall results still hold, but the difference between the methods is less stark than at more severe shifts. In particular, RLLS often does well at less severe shifts. This suggests possible extensions where the shift estimated by EM could be regularized, possibly by the use of a conjugate prior, so as to achieve better performance.
>
> “Optimizing ELBO with EM can lead to local convergence to the likelihood function when the likelihood is not unimodal. Is this likelihood function unimodal? Does the EM approach converges to MLE under some appropriate initialization and assumptions?”
> We have included a short proof in Appendix A showing that maximizing the likelihood under the relevant constraints is a convex optimization problem. Thus, we can anticipate EM will converge to the MLE.
>
> We have also combined some of the tables and expanded the captions to improve presentation.

---

### Decision · Program_Chairs · 2019-12-19

**Decision:**

Reject

**Comment:**

This was a borderline paper, but in the end two of the reviewers remain unconvinced by this paper in its current form, and the last reviewer is not willing to argue for acceptance. The first reviewer's comments were taken seriously in making a decision on this paper. As such, it is my suggestion that the authors revise the paper in its current form, and resubmit, addressing some of the first reviewers comments, such as discussion of utility of the methodology, and to improve the exposition such that less knowledgable reviewers understand the material presented better. The comments that the first reviewer makes about lack of motivation for parts of the presented methodology is reflected in the other reviewers comments, and I'm convinced that the authors can address this issue and make this a really awesome submission at a future conference.

On a different note, I think the authors should be congratulated on making their results reproducible. That is definitely something the field needs to see more of.